# coVariance Neural Networks

**Saurabh Sihag**
University of Pennsylvania
sihags@pennmedicine.upenn.edu

**Gonzalo Mateos**
University of Rochester
gmateosb@ece.rochester.edu

**Corey McMillan**
University of Pennsylvania
cmcmilla@pennmedicine.upenn.edu

**Alejandro Ribeiro**
University of Pennsylvania
aribeiro@seas.upenn.edu

## Abstract

Graph neural networks (GNN) are an effective framework that exploit inter-relationships within graph-structured data for learning. Principal component analysis (PCA) involves the projection of data on the eigenspace of the covariance matrix and draws similarities with the graph convolutional filters in GNNs. Motivated by this observation, we study a GNN architecture, called coVariance neural network (VNN), that operates on sample covariance matrices as graphs. We theoretically establish the stability of VNNs to perturbations in the covariance matrix, thus, implying an advantage over standard PCA-based data analysis approaches that are prone to instability due to principal components associated with close eigenvalues. Our experiments on real-world datasets validate our theoretical results and show that VNN performance is indeed more stable than PCA-based statistical approaches. Moreover, our experiments on multi-resolution datasets also demonstrate that VNNs are amenable to transferability of performance over covariance matrices of different dimensions; a feature that is infeasible for PCA-based approaches.

## 1 Introduction

Convolutional neural networks (CNN) are among the most popular deep learning (DL) architectures due to their demonstrated ability to learn latent features from Euclidean data adaptively and automatically [1]. Motivated by the fact that graph models can naturally represent data in many practical applications [2], the generalizations of CNNs to adapt to graph data using GNN architectures have rapidly emerged in recent years [3]. A practical challenge to DL architectures is scalability to high-dimensional data [4]. Specifically, DL models can typically have millions of parameters that require long training times and high storage capacity [5]. GNN architectures have two attractive features towards learning from high-dimensional data: 1. the number of parameters associated with graph convolution operations in GNNs is independent of the graph size; and 2. under certain assumptions, GNN performance can be transferred across graphs of different dimensions [6].

Principal component analysis (PCA) is a traditional, non-parametric approach for linear dimensionality reduction and increase of interpretability of high dimensional datasets [7]. Accordingly, it is common to embed the PCA transform within a DL pipeline [4, 8, 9]. Such approaches usually use PCA to optimize the DL model parameters by removing redundancy in the features in intermediate layers. The scope of our work is different. In this paper, we leverage the connections between PCA and the mathematical concepts used in GNNs to propose a learning framework that can recover PCA and also, inherit additional desirable features associated with GNNs.

36th Conference on Neural Information Processing Systems (NeurIPS 2022).

## 1.1 PCA and Graph Fourier Transform

PCA is an orthonormal transform which identifies the modes of maximum variance in the dataset. The transformed dimensions of the data (known as principal components) are uncorrelated. Computing the PCA transform reduces to finding the eigenbasis of the covariance matrix or equivalently, the singular value decomposition of the data matrix [10]. We note that PCA computation using eigendecomposition of a covariance matrix has similarities with the graph Fourier transform (GFT) in the graph signal processing literature [11]. This observation is discussed more formally next.

Consider an $m-$dimensional random vector $\mathbf{X} \in \mathbb{R}^{m \times 1}$ with the true or ensemble covariance matrix of $\mathbf{X}$ defined as $\mathbf{C} \triangleq \mathbb{E}[(\mathbf{X} - \mathbb{E}[\mathbf{X}])(\mathbf{X} - \mathbb{E}[\mathbf{X}])^\mathsf{T}]$. In practice, we may not observe the data model for $\mathbf{X}$ and its statistics, such as the covariance matrix $\mathbf{C}$ directly. Alternatively, we have access to a dataset consisting of $n$ random, independent and identically distributed (i.i.d) samples of $\mathbf{X}$, given by $\mathbf{x}_i \in \mathbb{R}^{m \times 1}, \forall i \in \{1, \ldots, n\}$, where the data matrix is $\hat{\mathbf{x}}_n = [\mathbf{x}_1, \ldots, \mathbf{x}_n]$. Using the samples $\mathbf{x}_i, \forall i \in \{1, \ldots, n\}$, we can form an estimate of the ensemble covariance matrix, conventionally referred to as the sample covariance matrix

$$\hat{\mathbf{C}}_n \triangleq \frac{1}{n} \sum_{i=1}^{n} (\mathbf{x}_i - \bar{\mathbf{x}})(\mathbf{x}_i - \bar{\mathbf{x}})^\mathsf{T} , \tag{1}$$

where $\bar{\mathbf{x}}$ is the sample mean across $n$ samples and $\cdot^\mathsf{T}$ refers to the transpose operator. Given the eigendecomposition of $\hat{\mathbf{C}}_n$

$$\hat{\mathbf{C}}_n = \mathbf{U}\mathbf{W}\mathbf{U}^\mathsf{T} , \tag{2}$$

where $\mathbf{U} = [\mathbf{u}_1, \ldots, \mathbf{u}_m]$ is the matrix with its columns as the eigenvectors and $\mathbf{W} = \mathrm{diag}(w_1, \ldots, w_m)$ is the diagonal matrix of eigenvalues, such that, $w_1 \geq w_2 \cdots \geq w_m$, the implementation of PCA is given by [10]

$$\hat{\mathbf{y}}_n = \mathbf{U}^\mathsf{T}\hat{\mathbf{x}}_n . \tag{3}$$

The eigenvectors of $\hat{\mathbf{C}}_n$ form the principal components and the transformation in (3) is equivalent to the projection of the sample $\mathbf{x}_i$ in the eigenvector space of $\hat{\mathbf{C}}_n$. A detailed overview of PCA is provided in Appendix B. We note that in the context of graph signal processing, (3) is equivalent to the graph Fourier transform (GFT) if the covariance matrix $\hat{\mathbf{C}}_n$ is considered the graph representation or a graph shift operator for the data $\hat{\mathbf{x}}_n$ [12].

## 1.2 Contributions

Motivated by the observation in Section 1.1, we introduce the notion of coVariance Fourier transform and define coVariance filters similar to graph convolutional filters in GNNs. Our analysis shows that standard PCA can be recovered using coVariance filters. Furthermore, we propose a DL architecture based on coVariance filters, referred to as coVariance neural networks (VNN)[1]: a permutation-invariant architecture derived from the GNN framework that uses convolutional filters defined on the covariance matrix. Understanding VNNs is relevant given the ubiquity of so-termed correlation networks [13, Chapter 7.3.1] and covariance matrices as graph models for neuroimaging [14], user preference [15], financial [16], and gene expression data[13, Chapter 7.3.4], to name a few timely domains. Summary of our contributions is as follows:

**Stability of VNN:** For finite $n$, the eigenvectors of the sample covariance matrix $\hat{\mathbf{C}}_n$ are perturbed from the eigenvectors of the ensemble covariance matrix $\mathbf{C}$. Moreover, the principal components corresponding to eigenvalues that are close are unstable, i.e., small changes in the dataset may cause large changes in such principal components [17] and subsequently, irreproducible statistical results [18]. Statistical learning approaches such as regression, classification, clustering that use PCA as the first step for dimension reduction inherit this instability. Our main theoretical contribution is to establish the *stability* of VNNs to perturbations in the sample covariance matrix in terms of sample size $n$. For this purpose, we leverage the perturbation theory for sample covariance matrices and show that for a given $\hat{\mathbf{C}}_n$, the perturbation in the VNN output $\Phi(\hat{\mathbf{C}}_n)$ with respect to the VNN output

---

[1]We emphasize on V in coVariance for abbreviations to avoid confusions with any existing machine learning frameworks.

for $\mathbf{C}$, $\Phi(\mathbf{C})$ satisfies $\|\Phi(\hat{\mathbf{C}}_n) - \Phi(\mathbf{C})\| = \mathcal{O}(1/n^{\frac{1}{2}-\varepsilon})$ for some $\varepsilon < 1/2$ with high probability. Therefore, our analysis ties the stability of VNNs to the quality of sample covariance matrices, which is distinct from existing studies that study stability of GNNs under abstract, data-independent discrete or geometric perturbations to graphs [19, 20].

**Empirical Validations:** Our experiments on real-world and synthetic datasets show that VNNs are indeed stable to perturbations in the sample covariance matrix. Moreover, on a multi-resolution dataset, we also illustrate the *transferability* of VNN performance to datasets of different dimensions without re-training. These observations provide convincing evidence for the advantages offered by VNNs over standard PCA-based approaches for data analysis.

### 1.3 Related Work

**GNNs**: GNNs broadly encapsulate all DL models adapted to graph data [3, 21]. Recent survey articles in [3] and [21] categorize the variants of GNNs according to diverse criteria that include mathematical formulations, algorithms, and hardware/software implementations. Convolutional GNNs [22], graph autoencoders [23], recurrent GNNs, and gated GNNs [24] are among a few prominently studied and applied categories of GNNs. The taxonomy pertinent to this paper is that of graph convolutional network (GCN) that generalizes the concept of convolution to graphs. Specifically, VNNs are based on GCNs that model the convolution operation via graph convolutional filters [25]. The graph convolutional filters are defined as polynomials over the graph representation where the order of the polynomial controls the aggregation of information over nodes in the graph [12]. We extend the definition of graph convolutional filters to filters over sample covariance matrices to define VNNs. The ubiquity of PCA and prevalence of covariance matrices across disciplines for data analysis motivate us to study VNNs independently of aforementioned GCNs. Moreover, the unique structure inherent to covariance matrices (relative to arbitrary graphs) and their sample-based perturbations facilitates sharper stability analyses, which offer novel insights.

**Robust PCA:** Robust PCA is a variant of PCA that aims to recover the low rank representation of a dataset arbitrarily corrupted due to outliers [26, 27, 28]. The outliers considered in robust PCA approaches could manifest in the dataset due to missing data, adversarial behavior, defects or contamination in data collection. In contrast to such settings, here we consider the inherent statistical uncertainty in the sample covariance matrix, where ill-defined eigenvalues and eigenvectors of the sample covariance matrix can be the source of instability in statistical inference. Therefore, the notion of stability of VNNs in this paper is fundamentally distinct from the notion of stability or robustness in robust PCA approaches.

## 2 coVariance Filters

If $m$ dimensions of data $\hat{\mathbf{x}}_n$ can be represented as the nodes of an $m$-node, undirected graph, the sample covariance matrix $\hat{\mathbf{C}}_n$ is equivalent to its adjacency matrix. In GNNs, graph Fourier transform projects graph signals in the eigenspace of the graph and is leveraged to analyze graph convolutional filters [12]. Therefore, we start by formalizing the notion of coVariance Fourier transform (abbreviated as VFT). For this purpose, we leverage the eigendecomposition of $\hat{\mathbf{C}}_n$ in (2).

**Definition 1** (coVariance Fourier Transform)**.** *Consider a sample covariance matrix $\hat{\mathbf{C}}_n$ as defined in (1). The coVariance Fourier transform (VFT) of a random sample $\mathbf{x}$ is defined as its projection on the eigenspace of $\hat{\mathbf{C}}_n$ and is given by*

$$\tilde{\mathbf{x}} \triangleq \mathbf{U}^\mathsf{T}\mathbf{x} \ . \tag{4}$$

The $i$-th entry of $\tilde{\mathbf{x}}$, i.e., $[\tilde{\mathbf{x}}]_i$ represents the $i$-th Fourier coefficient and is associated with the eigenvalue $w_i$. Note that the similarity between PCA and VFT implies that eigenvalue $w_i$ encodes the variability of dataset $\mathbf{x}_n$ in the direction of the principal component $\mathbf{u}_i$. In this context, the eigenvalues of the covariance matrix are mathematical equivalent of the notion of graph frequencies in graph signal processing [11]. Next, we define the notion of coVariance filters (VF) that are polynomials in the covariance matrix.

**Definition 2** (coVariance Filter). *Given a set of real valued parameters $\{h_k\}_{k=0}^m$, the coVariance filter for the covariance matrix $\hat{\mathbf{C}}_n$ is defined as*

$$\mathbf{H}(\hat{\mathbf{C}}_n) \triangleq \sum_{k=0}^m h_k \hat{\mathbf{C}}_n^k . \tag{5}$$

*The output of the covariance filter $\mathbf{H}(\hat{\mathbf{C}}_n)$ for an input $\mathbf{x}$ is given by*

$$\mathbf{z} = \sum_{k=0}^m h_k \hat{\mathbf{C}}_n^k \mathbf{x} = \mathbf{H}(\hat{\mathbf{C}}_n)\mathbf{x} . \tag{6}$$

The coVariance filter $\mathbf{H}(\hat{\mathbf{C}}_n)$ follows similar analytic concepts of combining information in different neighborhoods as in the well-studied graph convolutional filters [12]. Moreover, the filter is defined by the parameters $\{h_k\}_{k=0}^m$ and therefore, for the ensemble covariance matrix $\mathbf{C}$, the coVariance filter is given by $\mathbf{H}(\mathbf{C})$. On taking the VFT of the output in (6) and leveraging (2), we have

$$\tilde{\mathbf{z}} \triangleq \mathbf{U}^\mathsf{T}\mathbf{z} = \mathbf{U}^\mathsf{T} \sum_{k=0}^m h_k [\mathbf{U}\mathbf{W}\mathbf{U}^\mathsf{T}]^k \mathbf{x} = \sum_{k=0}^m h_k \mathbf{W}^k \tilde{\mathbf{x}} , \tag{7}$$

where (7) holds from the orthonormality of eigenvectors and definition of VFT in (4). Using (7), we can further define the frequency response of the coVariance filter over the covariance matrix $\hat{\mathbf{C}}_n$ in the domain of its principal components as

$$h(w_i) = \sum_{k=0}^m h_k w_i^k , \tag{8}$$

such that, from (7) and (8), the $i$-th element of $\tilde{\mathbf{z}}_i$ has the following relationship

$$[\tilde{\mathbf{z}}]_i = h(w_i)[\tilde{\mathbf{x}}]_i . \tag{9}$$

Equation (9) reveals that performing the coVariance filtering operation boils down to processing (e.g., amplifying or attenuating) the principal components of the data. This observation draws analogy with the linear-time invariant systems in signal processing where the different frequency modes (in this case, principal components) can be processed separately using coVariance filters, in a way determined by the frequency response values $h(w_i)$. For instance, using a narrowband coVariance filter whose frequency response is $h(\lambda) = 1$, if $\lambda = w_i$ and $h(\lambda) = 0$, otherwise, we recover the score corresponding to the projection of $\mathbf{x}$ on $\mathbf{u}_i$, i.e, the $i$-th principal component of $\hat{\mathbf{C}}_n$. Therefore, there exist filterbanks of narrowband coVariance filters that enable the recovery of the PCA transformation. This observation is formalized in Theorem 1.

**Theorem 1** (coVariance Filter Implementation of PCA). *Given a covariance matrix $\hat{\mathbf{C}}_n$ with eigen-decomposition in (2), if the PCA transformation of input $\mathbf{x}$ is given by $\mathbf{y} = \mathbf{U}^\mathsf{T}\mathbf{x}$, there exists a filterbank of coVariance filters $\{\mathbf{H}_i(\hat{\mathbf{C}}_n) : i \in \{1, \ldots, m\}\}$, such that, the score of the projection of input $\mathbf{x}$ on eigenvector $\mathbf{u}_i$ can be recovered by the application of a coVariance filter $\mathbf{H}_i(\hat{\mathbf{C}}_n)$ as:*

$$[\mathbf{y}]_i = \mathbf{u}_i^\mathsf{T} \mathbf{H}_i(\hat{\mathbf{C}}_n)\mathbf{x} , \tag{10}$$

*where the frequency response $h_i(\lambda)$ of the filter $\mathbf{H}_i(\hat{\mathbf{C}}_n)$ is given by*

$$h_i(\lambda) = \begin{cases} \eta_i, & if \quad \lambda = w_i , \\ 0, & otherwise \end{cases} . \tag{11}$$

Theorem 1 establishes equivalence between processing data samples with PCA and processing data samples with a specific polynomial on the covariance matrix. As we shall see in subsequent sections, the processing on a polynomial of covariance matrix has advantages in terms of stability with respect to the perturbations in the sample covariance matrix. The design of frequency response of different coVariance filters sufficient to implement PCA transformation according to Theorem 1 is shown figuratively in Appendix D. If it is desired to have PCA-based data transformation be followed by dimensionality reduction or statistical learning tasks such as regression or classification, the coVariance filter-based PCA can be coupled with the post-hoc analysis model to have an end-to-end framework that enables the optimization of the parameters $\eta_i$ in the frequency response of the coVariance filters.

# 3   coVariance Neural Networks (VNN)

In this section, we propose coVariance Neural Network (VNN), which provides an end-to-end, non-linear, parametric mapping from the input data $\mathbf{x}$ to any generic objective $\mathbf{r}$ and is defined as

$$\mathbf{r} = \Phi(\mathbf{x}; \hat{\mathbf{C}}_n, \mathcal{H}) \,, \tag{12}$$

for sample covariance matrix $\hat{\mathbf{C}}_n$ where $\mathcal{H}$ is the set of filter coefficients that characterize the representation space defined by the mapping $\Phi(\cdot)$. The VNN $\Phi(\mathbf{x}; \hat{\mathbf{C}}_n, \mathcal{H})$ may be formed by multiple layers, where each layer consists of two main components: i) a filter bank made of VFs similar to that in (6); and ii) a pointwise non-linear activation function (such as ReLU, $\tanh$). Therefore, in principle, the architecture of VNNs is similar to that of graph neural networks with the covariance matrix $\hat{\mathbf{C}}_n$ as the graph shift operator [12]. We next define the coVariance perceptron, which forms the building block of a VNN and is equivalent to a 1-layer VNN.

**Definition 3** (coVariance Perceptron). *Consider a dataset with the sample covariance matrix $\hat{\mathbf{C}}_n$. For a given non-linear activation function $\sigma(\cdot)$, input $\mathbf{x}$, a coVariance filter $\mathbf{H}(\hat{\mathbf{C}}_n) = \sum_{k=0}^{m} h_k \hat{\mathbf{C}}_n^k$ and its corresponding coefficient set $\mathcal{H}$, the coVariance perceptron is defined as*

$$\Phi(\mathbf{x}; \hat{\mathbf{C}}_n, \mathcal{H}) \triangleq \sigma(\mathbf{H}(\hat{\mathbf{C}}_n)\mathbf{x}) \,. \tag{13}$$

The VNN can be constructed by cascading multiple layers of coVariance perceptrons (shown in Appendix D in the Supplementary file). Note that the non-linear activation functions across different layers allow for non-linear transformations, thus, increasing the expressiveness of VNNs beyond linear mappings such as coVariance filters. Furthermore, similar to GNNs, we can further increase the representation power of VNNs by incorporating multiple parallel inputs and outputs per layer enabled by filter banks at every layer [12, 29]. In this context, we remark that the definition of a one layer perceptron in (13) can be expanded to the following.

**Remark 1** (coVariance Perceptron with Filter Banks). *Consider a coVariance perceptron with $F_{\text{in}}$ $m$-dimensional inputs and $F_{\text{out}}$ $m$-dimensional outputs. Denote the input at the perceptron by $\mathbf{x}_{\text{in}} = \left[\mathbf{x}_{\text{in}}[1], \ldots, \mathbf{x}_{\text{in}}[F_{\text{in}}]\right]$ and the output at the perceptron by $\mathbf{x}_{\text{out}} = \left[\mathbf{x}_{\text{out}}[1], \ldots, \mathbf{x}_{\text{out}}[F_{\text{out}}]\right]$. Each input $\mathbf{x}_{\text{in}}[g], \forall g \in \{1, \ldots, F_{\text{in}}\}$ is processed by $F_{\text{out}}$ coVariance filters in parallel. For $f \in \{1, \ldots, F_{\text{out}}\}$, the $f$-th output in $\mathbf{x}_{\text{out}}$ is given by*

$$\mathbf{x}_{\text{out}}[f] = \sigma\left(\sum_{g=1}^{F_{\text{in}}} \mathbf{H}_{fg}(\hat{\mathbf{C}}_n)\mathbf{x}_{\text{in}}[g]\right) = \Phi(\mathbf{x}_{\text{in}}; \hat{\mathbf{C}}_n, \mathcal{H}_f) \,, \tag{14}$$

*where $\mathcal{H}_f$ is the set of all filter coefficients in coVariance filters $[\mathbf{H}_{fg}(\hat{\mathbf{C}}_n)]_g$ in (14).*

Therefore, the VNN with filter bank implementation deploys $F_{\text{in}} \times F_{\text{out}}$ number of VFs in a layer defined by the covariance perceptron in Remark 1. Next, we note that the definitions of coVariance filter in Section 2 and VNN are with respect to the sample covariance matrix $\hat{\mathbf{C}}_n$. However, due to finite sample size effects, the sample covariance matrix will be a perturbed version of the ensemble covariance matrix $\mathbf{C}$. PCA-based approaches that rely on eigendecomposition of the sample covariance matrix can potentially be highly sensitive to such perturbations. Specifically, if small changes are made to the dataset $\mathbf{x}_n$, certain ill-defined eigenvalues and eigenvectors can induce instability in the outcomes of PCA-based statistical learning models [17]. Therefore, it is desirable that the designs of coVariance filters and VNNs are not sensitive to random perturbations in the sample covariance matrix as compared to the ensemble covariance matrix $\mathbf{C}$. In this context, we study the stability of coVariance filters and VNNs in the next section.

## 4   Stability Analysis

We start with the stability analysis of coVariance filters, which will also be instrumental in establishing the stability of VNNs. Our results will leverage the eigendecomposition of $\mathbf{C}$, given by

$$\mathbf{C} = \mathbf{V}\Lambda\mathbf{V}^{\mathsf{T}} \tag{15}$$

where $\mathbf{V} \in \mathbb{R}^{m \times m}$ is the matrix of eigenvectors such that, $\mathbf{V} = [\mathbf{v}_1, \ldots, \mathbf{v}_m]$ and $\Lambda = \text{diag}(\lambda_1, \ldots, \lambda_m)$ is the diagonal matrix of eigenvalues, such that, $\lambda_1 \geq \cdots \geq \lambda_m$.

### 4.1 Stability of coVariance Filters

To study the stability of coVariance filters, we analyze the effect of statistical uncertainty induced in the sample covariance matrix with respect to the ensemble covariance matrix on the coVariance filter output. To this end, we compare the outputs $\mathbf{H}(\mathbf{C})\mathbf{x}$ and $\mathbf{H}(\hat{\mathbf{C}}_n)\mathbf{x}$ for any random instance $\mathbf{x}$ of $\mathbf{X}$. Without loss of generality, we present our results over instances of $\mathbf{X}$ where $\|\mathbf{X}\| \leq 1$. We also consider the following assumption on the frequency response.

**Assumption**: The frequency response of filter $\mathbf{H}(\mathbf{C})$ satisfies:

$$|h(\lambda_i) - h(\lambda_j)| \leq M \frac{|\lambda_i - \lambda_j|}{k_i} \, , \tag{16}$$

where $k_i \triangleq \sqrt{\mathbb{E}[\|\mathbf{X}\mathbf{X}^\mathsf{T}\mathbf{v}_i\|^2] - \lambda_i^2}$, for some constant $M > 0$ and for all non-zero eigenvalues $\lambda_i \neq \lambda_j, i, j \in \{1, \dots, m\}$ of $\mathbf{C}$. Here, $k_i$ is a measure of kurtosis of the distribution of $\mathbf{X}$.

Next, we provide the result that establishes the stability of coVariance filters in Theorem 2. The impact of the assumption in (16) on the filter stability is discussed subsequently. To present the results in Theorem 2, we also use the following definitions:

$$k_{\mathsf{min}} \triangleq \min_{i \in \{1, \dots, m\}, \lambda_i > 0} k_i \quad \text{and} \quad \kappa \triangleq \max_{i,j:\lambda_i \neq \lambda_j} \frac{k_i^2}{|\lambda_i - \lambda_j|} \, . \tag{17}$$

**Theorem 2** (Stability of coVariance Filter). *Consider a random vector* $\mathbf{X} \in \mathbb{R}^{m \times 1}$ *, such that, its corresponding covariance matrix is given by* $\mathbf{C} = \mathbb{E}[(\mathbf{X} - \mathbb{E}[\mathbf{X}])(\mathbf{X} - \mathbb{E}[\mathbf{X}])^\mathsf{T}]$. *For a sample covariance matrix* $\hat{\mathbf{C}}_n$ *formed using* $n$ *i.i.d instances of* $\mathbf{X}$ *and a random instance* $\mathbf{x}$ *of* $\mathbf{X}$, *such that,* $\|\mathbf{x}\| \leq 1$ *and under assumption* (16), *the following holds with probability at least* $1 - n^{-2\varepsilon} - 2\kappa m/n$ *for any* $\varepsilon \in (0, 1/2]$:

$$\left\| \mathbf{H}(\hat{\mathbf{C}}_n) - \mathbf{H}(\mathbf{C}) \right\| = \frac{M}{n^{\frac{1}{2}-\varepsilon}} \cdot \mathcal{O}\left( \sqrt{m} + \frac{\|\mathbf{C}\|\sqrt{\log mn}}{k_{\mathsf{min}} n^{2\varepsilon}} \right) \, . \tag{18}$$

*Proof.* See Appendix C. □

The right-hand side term in (18) and the conditions in the assumption in (16) are obtained by the analysis of the finite sample size effect-driven perturbations in $\hat{\mathbf{C}}_n$ and its eigenvectors with respect to that in $\mathbf{C}$. From Theorem 2, we note that $\|\mathbf{H}(\hat{\mathbf{C}}_n) - \mathbf{H}(\mathbf{C})\|$ decays with the number of samples $n$ at least at the rate of $1/n^{\frac{1}{2}-\varepsilon}$. Thus, we conclude that the stability of the coVariance filter improves as the number of samples $n$ increases. This observation is along the expected lines as the estimate $\hat{\mathbf{C}}_n$ becomes closer to the ensemble covariance matrix $\mathbf{C}$ by the virtue of the law of large numbers. Next, we briefly discuss two aspects of the assumption in (16). Note that the upper bound in (16) controls the variability of the frequency response $h(\lambda)$ with respect to $\lambda$. For any pair of eigenvalues $\lambda_i$ and $\lambda_j$ of $\mathbf{C}$, this variability is tied to the eigengap $|\lambda_i - \lambda_j|$ and the factor $k_i$. We discuss this next.

**Discriminability between close eigenvalues**: Firstly, for a given $k_i$ and eigenvalue $\lambda_i$, the response of the filter for any eigenvalue $\lambda_j, j \neq i$ becomes closer to $h(\lambda_i)$ if $|\lambda_i - \lambda_j|$ decreases. From the perturbation theory of eigenvectors and eigenvalues of sample covariance matrices, we know that the sample-based estimates of the eigenspaces corresponding to eigenvalues $\lambda_i$ and $\lambda_j$ become harder to distinguish as $|\lambda_i - \lambda_j|$ decreases [30]. Therefore, the coVariance filter that satisfies (16) sacrifices discriminability between close eigenvalues to preserve its stability with respect to the statistical uncertainty inherent in the sample covariance matrix.

**Stability with respect to kurtosis and estimation quality**: Since we have $\mathbb{E}[\|\mathbf{X}\mathbf{X}^\mathsf{T}\mathbf{v}_i\|^2] \leq \mathbb{E}[\|\mathbf{X}\|^4]$, the factor $k_i$ is tied to the measure of kurtosis of the underlying distribution of $\mathbf{X}$ in the direction of $\mathbf{v}_i$. Distributions with high kurtosis tend to have heavier tails and more outliers. Therefore, smaller $k_i$ indicates that the distribution of $\mathbf{X}$ has a fast decaying tail in the direction of $\mathbf{v}_i$ which allows for a more accurate estimation of $\lambda_i$ and $\mathbf{v}_i$. We refer the reader to [30, Section 4.1.3] for additional details. In the context of coVariance filters, we note that the upper bound in (16) is more liberal if $\lambda_i$ is associated with a smaller $k_i$ or equivalently, the distribution of $\mathbf{X}$ has a smaller kurtosis in the direction of $\mathbf{v}_i$. This observation implies that if $\lambda_i$ and $\mathbf{v}_i$ are 'easier' to estimate, the frequency response for $\lambda_j$ in the vicinity of $\lambda_i$ is less constrained.

## 4.2 Stability of coVariance Neural Networks

The stability of VNNs is analyzed by comparing $\Phi(\mathbf{x}; \hat{\mathbf{C}}_n, \mathcal{H})$ and $\Phi(\mathbf{x}; \mathbf{C}, \mathcal{H})$. Note that stable graph convolutional filters imply the stability of GNNs for different perturbation models [19]. In VNNs, the perturbations are derived from the finite sample effect in sample covariance matrix, which is distinct from the data-independent perturbation models considered in the existing literature on GNNs, and allow us to relate sample size $n$ with VNN stability. We formalize the stability of VNNs under the assumption of stable coVariance filters in Theorem 3. For this purpose, we consider a VNN $\Phi(\cdot; \cdot, \mathcal{H})$ with number of $m$-dimensional inputs and outputs per layer as $F$ and $L$ layers, with the filter bank given by $\mathcal{H} = \{\mathbf{H}^\ell_{fg}\}, \forall f, g \in \{1, \dots, F\}, \ell \in \{1, \dots, L\}$.

**Theorem 3** (Stability of VNN). *Consider a sample covariance matrix $\hat{\mathbf{C}}_n$ and the ensemble covariance matrix $\mathbf{C}$. Given a bank of coVariance filters $\{\mathbf{H}^\ell_{fg}\}$, such that $|h^\ell_{fg}(\lambda)| \leq 1$ and a pointwise non-linearity $\sigma(\cdot)$, such that, $|\sigma(a) - \sigma(b)| \leq |a - b|$, if the covariance filters satisfy*

$$\|\mathbf{H}^\ell_{fg}(\hat{\mathbf{C}}_n) - \mathbf{H}^\ell_{fg}(\mathbf{C})\| \leq \alpha_n \, , \tag{19}$$

*for some $\alpha_n > 0$, then, we have*

$$\|\Phi(\mathbf{x}; \hat{\mathbf{C}}_n, \mathcal{H}) - \Phi(\mathbf{x}; \mathbf{C}, \mathcal{H})\| \leq LF^{L-1}\alpha_n \, . \tag{20}$$

The proof of Theorem 3 follows from [12, Appendix E] for any generic $\alpha_n$. The parameter $\alpha_n$ represents the finite sample effect on the perturbations in $\hat{\mathbf{C}}_n$ with respect to $\mathbf{C}$. From Theorem 2, we note that $\alpha_n$ scales as $1/n^{\frac{1}{2}-\varepsilon}$ with respect to the number of samples $n$ for the coVariance filters whose frequency response depends on the eigengap and kurtosis of the underlying distribution of the data in (16). Furthermore, the stability of a VNN decreases with increase in number of $m$-dimensional inputs and outputs per layer $F$ and number of layers $L$, which is consistent with the stability properties of GNNs. Therefore, our results present a more holistic perspective to the stability of VNNs than that possible for generic GNNs.

**Remark 2** (Computational Complexity of VNN). *For a coVariance perceptron defined in (14), the computational cost is given by $O(m^2 T F_{\mathsf{in}} F_{\mathsf{out}})$, where $T \leq m$ is the maximum number of filter taps in its associated filter bank.*

From Remark 2, we note that the computational complexity of VNNs can be prohibitive for large $m$. However, oftentimes sparsity is imposed as a regularization to estimate high-dimensional correlation matrices; see e.g. [31]. As a result, the computational complexity becomes $O(|E|T F_{\mathsf{in}} F_{\mathsf{out}})$, where $|E|$ is the number of non-zero correlations (edges in the covariance graph) and can be markedly smaller than $m^2$. Since VNN architecture is analogous to that of GNN, the property of GNN transferability (see [6]) across different sized graphs can also establish scalability of VNNs to high-dimensional datasets for settings where multi-resolution datasets are available. In the next section, we empirically validate our theoretical results on the stability of VNNs. Moreover, on a set of multi-resolution datasets, we also empirically evaluate VNNs for transferability.

## 5 Experiments

In this section, we discuss our experiments on different datasets. Primarily, we evaluate VNNs on a regression problem on different neuroimaging datasets curated at University of Pennsylvania, where we regress human chronological age (time since birth) against cortical thickness data. Cortical thickness is a measure of the gray matter width and it is well established that cortical thickness varies with healthy aging [32]. Additional details on the neurological utility of this experiment are included in Appendix E. Brief descriptions of these datasets are provided next.

**ABC Dataset**: ABC dataset consists of the cortical thickness data collected from a heterogeneous population of $n = 341$ subjects (mean age = 68.1 years, standard deviation = 13) that consists of healthy adults, and subjects with mild cognitive impairment or Alzheimer's disease. For each individual, joint-label fusion [33] was used to quantify mean cortical thickness in $m = 104$ anatomical parcellations. Therefore, for every subject, we have a 104 dimensional vector whose entries correspond to the cortical thickness in distinct brain regions.

**Multi-resolution FTDC Datasets**: FTDC Datasets consist of the cortical thickness data from $n = 170$ healthy subjects (mean age = 64.26 years, standard deviation = 8.26). For each subject,

the cortical thickness data is extracted according to a multiresolution Schaefer parcellation [34], at 100 parcel, 300 parcel, and 500 parcel resolutions. Therefore, for each subject, we have the cortical thickness data consisting of $m = 100$ features, $m = 300$ features or $m = 500$ features, with the higher number of features providing the cortical thickness data of a brain at a finer resolution. We leverage the different resolutions of data available to form three datasets: FTDC100, FTDC300, and FTDC500, which form the cortical thickness datasets corresponding to 100, 300, and 500 features resolutions, respectively.

Our primary objective is to illustrate the stability and transferability of VNNs that also imply advantages over traditional PCA-based approaches. Hence, we use PCA-based regression as the primary baseline for comparison against VNNs. We use the ABC dataset to demonstrate the higher stability of VNNs over PCA-based regression models in Section 5.1. In Section 5.2, we use the FTDC datasets to demonstrate transferability of VNN performance across the multi-resolution datasets without re-training. The experiments in Section 5.2 clearly lay beyond the scope of PCA-based statistical learning models. Details on data and code availability for the experiments are included in Appendix A.

## 5.1 Stability against Perturbations in Sample Covariance Matrix

In this section, we evaluate the stability or robustness of the trained VNN and PCA-regression models against perturbations in the sample covariance matrix used in training. To this end, we first train nominal VNN and PCA-regression models. The effects of perturbations in the sample covariance matrix on the performance of nominal models are evaluated subsequently. We first describe the experiment designs for nominal models based on VNN and PCA-regression.

**VNN Experiments**: We randomly split ABC dataset into a $90/10$ train/test split. The sample covariance matrix is formed from 307 samples in the training set, i.e., we have $\hat{\mathbf{C}}_{307}$ of size $104 \times 104$. The VNN consists of 2 layers with 2 filter taps each, a filter bank of 13 $m$-dimensional outputs per layer for $m = 104$ dimensions of the input data, and a readout layer that calculates the unweighted mean of the outputs at the last layer to form an estimate for age. The hyperparameters for the VNN architecture and learning rate of the optimizer in this experiments and all subsequent VNN experiments in this section are chosen by a hyperparameter optimization framework called Optuna [35]. The training set is randomly subdivided into subsets of 273 and 34 samples, such that, the VNN is trained with respect to the mean squared error loss between the predicted age and the true age in 273 samples. The loss is optimized using batch stochastic gradient descent with Adam optimizer available in PyTorch library [36] for up to 100 epochs. The learning rate for the optimizer is set to $0.0151$. The VNN model with the best minimum mean squared error performance on the remaining 34 samples (which acts as a validation set) is included in the set of nominal models for this permutation of the training set.

**PCA-regression**: The PCA-regression pipeline consists of two steps: i) we first identify the principle components using the eigendecomposition of the sample covariance matrix $\hat{\mathbf{C}}_{307}$; and then, ii) to maintain consistency with VNN, transform the 273 samples from the training set used for VNN training to fit to the corresponding age data using a regression model. Regression is implemented using sklearn package in python [37] with 'linear' and radial basis function ('rbf') kernels. PCA-regression with 'rbf' kernel enables us to accommodate non-linear relationships between cortical thickness and age. PCA-regression with linear kernel in the regression model is referred to as PCA-LR and that with 'rbf' kernel in the regression model is referred to as PCA-rbf. The optimal number of principal components in the PCA-regression pipeline are selected through a 10-fold cross-validation procedure on the training set, repeated 5 times.

For 100 random permutations of the training set in VNN and PCA-regression experiments, we form a set of 100 nominal models and evaluate their stability. To evaluate the stability of VNN, we replace the sample covariance matrix $\hat{\mathbf{C}}_{307}$ with $\hat{\mathbf{C}}_{n'}$ for $n' \in [5, 341], n' \neq 307$. For PCA-regression models, we re-evaluate the principal components corresponding to $\hat{\mathbf{C}}_{n'}$ to transform the training data while keeping the regression model learnt for PCA transformation from $\hat{\mathbf{C}}_{307}$ fixed. Clearly, $\hat{\mathbf{C}}_{n'}$ will be perturbed with respect to $\hat{\mathbf{C}}_{307}$ due to finite sample size effect.

For each nominal model, we evaluate the model performances in terms of mean absolute error (MAE) and correlation between predicted age and true age for the training set and the test set.

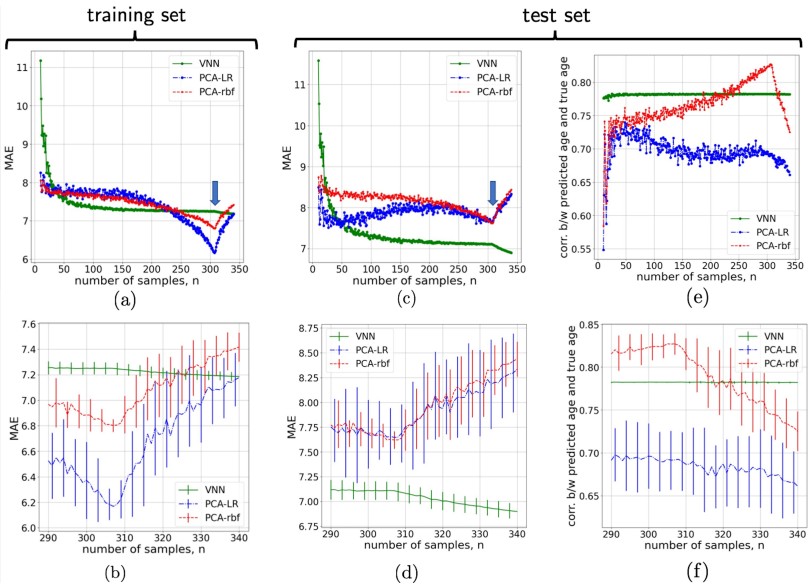

Figure 1: Stability of regression performance for VNN and PCA-regression models trained on $\hat{\mathbf{C}}_{307}$ formed from ABC dataset. Panels (a) and (b) illustrate the performance variation on training set for the VNN and PCA-regression models when sample covariance matrix $\hat{\mathbf{C}}_{307}$ is perturbed by addition or removal of samples ($n = 307$ marked by blue arrow in (a)). Panels (c)-(f) correspond to the variation in performances in terms of MAE and correlation between predicted age and true age only on the test set. The point $n = 307$ is marked with a blue arrow in panel (c). Panels (c) and (d) illustrate the variation in mean MAE performance of the VNN and PCA-regression models trained using $\hat{\mathbf{C}}_{307}$ on the test set ($n = 307$ marked by blue arrow in (c)) with panel (d) zooming in on the range $290 - 341$ with error bars included. Panel (e) illustrates the variation in correlation between true age and predicted age by the VNN and PCA-regression models with panel (f) zooming in on the range $290 - 341$ with error bars included.

Figures 1 a)-b) illustrate the variation in MAE performance with perturbations to $\hat{\mathbf{C}}_{307}$ on the training set. Figures 1 c)-f) correspond to similar results on the test set for MAE and correlation. The mean performance in terms of MAE over 100 nominal models is marked by a blue arrow in Fig. 1 a) for training set and in Fig. 1 c) for the test set. For PCA-regression models, we observe that both training and test performance in terms of MAE degrades significantly when the sample covariance matrix is perturbed from $\hat{\mathbf{C}}_{307}$ by removing or adding even a small number of samples (also seen in Fig. 1 b) and d), which are the plots focused only on the range of $290 - 341$ samples from Fig. 1 a) and c), respectively, and include error bars.). In contrast, the VNN performances on both training and test sets are stable with respect to perturbations to $\hat{\mathbf{C}}_{307}$, as suggested by our theoretical results. However, as the number of samples decrease to $n' < 50$, we observe a significant decrease in VNN stability in Fig. 1 a) and c). We also observe that the correlation between the predicted age and true age in the test set for VNN is consistently more stable than that for PCA-regression models over the entire range of samples evaluated (Fig. 1 e)). Moreover, Fig. 1 e) also demonstrates that in the range of $n' < 50$ samples, there is a sharper decline in the correlation for PCA-regression models as compared to VNN despite the MAE for PCA-regression models having smaller MAE than VNN in this range. Thus, our experiments demonstrate the stability of VNNs while also illustrating that PCA-regression models may be overfit on the principal components of the sample covariance matrix used in training. Additional experiments on synthetic data and FTDC datasets also illustrate the stability of VNNs (see Appendix E).

## 5.2 Transferability

We evaluate the transferability for VNN models trained on FTDC datasets across different resolutions. For this purpose, the VNN is trained at a specific resolution and its sample covariance matrix is replaced by the sample covariance matrix at a different resolution for testing. The training of VNNs

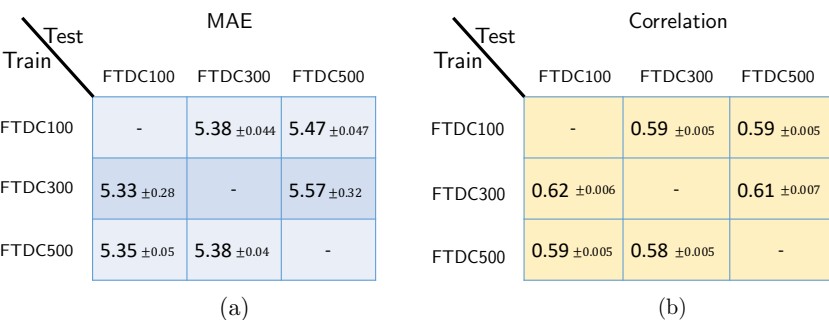

Figure 2: Transferability for VNNs across FTDC datasets.

in this context follows similar procedure as described in Section 5.1 and results in 100 random VNN models for different permutations of the training set. For FTDC500, the VNN model consists of 1-layer with a filter bank of 27 $m$-dimensional outputs per layer for $m = 500$ dimensions of the input data, and 2 filter taps in each layer. The learning rate for the Adam optimizer is set to $0.008$. For FTDC300, the VNN model consists of 2-layer architecture, with $44$ $m$-dimensional outputs per layer for $m = 300$ dimensions in both layers and 2 filter taps in each layer. The learning rate for the Adam optimizer is set to $0.0033$. For FTDC100, the VNN model consists of 2-layer architecture, with $93$ $m$-dimensional outputs per layer for $m = 100$ dimensions in both layers and 2 filter taps in each layer. The learning rate for the optimizer is set to $0.0047$. The readout layer in each model evaluates the unweighted mean of the outputs of the final layer to form an estimate for age.

We tabulate the MAE in the matrix in Fig. 2 a) and correlation between true age and predicted age in the matrix in Fig. 2 b), where the row ID indicates the dataset for which VNN was trained and the column ID indicates the dataset on which the VNN was tested to evaluate transferability of VNN performance across datasets with different resolutions. For instance, the element at coordinates $(1, 2)$ in Fig. 2a) represents the MAE evaluated on complete FTDC300 dataset ($m = 300$) for VNNs trained on FTDC100 dataset ($m = 100$). The results in Fig. 2 show that the performance of VNNs in terms of MAE and correlation between predicted age and true age can be transferred across different resolutions in the FTDC datasets. Note that this experiment is not feasible for PCA-regression models, where the principal components and the regression model would need to be re-evaluated for data from different resolution.

## 6   Conclusions

In this paper, we have introduced coVariance neural networks (VNN) that operate on covariance matrices and whose architecture is derived from graph neural networks. We have studied the stability properties of VNNs for sample covariance matrices and shown that the stability of VNNs improves with the number of data samples $n$. Our experiments on real datasets have demonstrated that VNNs are significantly more stable than PCA-based approaches with respect to perturbations in the sample covariance matrix. Also, unlike PCA-based approaches, VNNs do not require the eigendecomposition of the sample covariance matrix. Furthermore, on a set of multiresolution datasets, we have observed that VNN performance is also transferable across cortical thickness data collected at multiple resolutions without re-training.

## Acknowledgements

This research was supported by National Science Foundation under the grants CCF-1750428 and CCF-1934962. The ABC dataset was provided by the Penn Alzheimer's Disease Research Center (ADRC; NIH AG072979) at University of Pennsylvania. The MRI data for FTDC datasets were provided by the Penn Frontotemporal Degeneration Center (NIH AG066597). Cortical thickness data were made available by Penn Image Computing and Science Lab at University of Pennsylvania.

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
