## A  Data and Code Availability

Data for experiments on ABC and FTDC datasets may be requested through `https://www.pennbindlab.com/data-sharing` and upon review by the University of Pennsylvania Neurodegenerative Data Sharing Committee, access will be granted upon reasonable request. Data and code for synthetic experiments in Appendix D are available at `https://github.com/pennbindlab/VNN`. The implementations of the experiments on ABC and FTDC datasets are similar.

## B  Overview of PCA

The advantages of using PCA, its role in dimension reduction, and limitations are well documented [10]. PCA is an orthogonal, linear transformation of the observed dataset $\hat{\mathbf{x}}_n$ via a change of basis, such that, the transformed data has minimal redundancy while the hidden structure or patterns of interest in the raw data are preserved for further statistical analyses. In practice, the implementation of PCA is given by

$$\hat{\mathbf{y}}_n = \mathbf{P}\hat{\mathbf{x}}_n \ , \tag{21}$$

where the rows of $\mathbf{P} \in \mathbb{R}^{m \times m}$ represent the new basis on which the data $\hat{\mathbf{x}}_n$ is projected and therefore, form the designed principal components in PCA. We next briefly discuss the construction of the transformation $\mathbf{P}$ and the intuition behind it. Note that in the diagonal elements of the sample covariance matrix $\hat{\mathbf{C}}_n$ represent the individual variances of the constituent $m$ features and its off-diagonal elements characterize the redundancy in the data. A desired objective of PCA is to de-correlate the dataset $\hat{\mathbf{x}}_n$ by removing any second order dependencies. To achieve this, a linear transformation $\mathbf{P}$ is designed, such that, the covariance of the transformed data $\mathbf{y}_n$, given by $\hat{\mathbf{C}}_\mathbf{y}$ is a diagonal matrix. It can readily be verified that the eigenvectors of the sample covariance matrix $\hat{\mathbf{C}}_n$ satisfy this property. The eigendecomposition of $\hat{\mathbf{C}}_n$ is formalized as

$$\hat{\mathbf{C}}_n = \mathbf{U}\mathbf{W}\mathbf{U}^\mathsf{T} \ , \tag{22}$$

where $\cdot^\mathsf{T}$ refers to the Hermitian operator, $\mathbf{U}$ is the matrix constituted by orthonormal eigenvectors of $\hat{\mathbf{C}}_n$ and $\mathbf{W} = \mathrm{diag}(w_1, \ldots, w_m)$ is the diagonal matrix of eigenvalues of $\hat{\mathbf{C}}_n$, such that, $w_1 \geq w_2 \cdots \geq w_m$. Therefore, for transformation $\mathbf{P} = \mathbf{U}^\mathsf{T}$, such that, $\hat{\mathbf{y}}_n = \mathbf{U}^\mathsf{T}\hat{\mathbf{x}}_n$ we have

$$\hat{\mathbf{C}}_\mathbf{y} = \frac{1}{n}\mathbf{U}^\mathsf{T}\mathbf{x}_n\mathbf{x}_n^\mathsf{T}\mathbf{U} = \mathbf{U}^\mathsf{T}(\mathbf{U}\mathbf{W}\mathbf{U}^\mathsf{T})\mathbf{U} = \mathbf{W} \ . \tag{23}$$

Hence, in practical implementation of PCA, the eigenvectors of the sample covariance matrix form the principal components that characterize the linear, orthogonal transformation of data $\hat{\mathbf{x}}_n$. A common application of PCA is to extract the principal components that explain the most variance in $\hat{\mathbf{x}}_n$ (thus, resulting in dimension reduction while preserving the most relevant information under certain assumptions), followed by designing statistical models for inference tasks such as prediction, classification etc [7].

## C  Stability of coVariance Filters

We start by characterizing the perturbation of sample covariance matrix $\hat{\mathbf{C}}_n$ with respect to $\mathbf{C}$ in Lemma 1. For convenience, we use the notation $\hat{\mathbf{C}}$ for $\hat{\mathbf{C}}_n$. To this end, we define

$$\mathbf{E} \triangleq \hat{\mathbf{C}} - \mathbf{C} \ , \tag{24}$$

and $\mathbf{I}_m$ as an $m \times m$ identity matrix.

**Lemma 1.** *Consider an ensemble covariance matrix $\mathbf{C}$ with the eigendecomposition in (15) and a sample covariance matrix $\hat{\mathbf{C}}$ with the eigendecomposition in (1). For any eigenvalue $\lambda_i > 0$ of $\mathbf{C}$, the perturbation $\mathbf{E}$ satisfies*

$$\mathbf{E}\mathbf{v}_i = \beta_i\delta\mathbf{v}_i + \delta\lambda_i\mathbf{v}_i + (\delta\lambda_i\mathbf{I}_m - \mathbf{E})\delta\mathbf{v}_i \tag{25}$$

*where*

$$\beta_i \triangleq (\lambda_i\mathbf{I}_m - \mathbf{C}), \quad \delta\mathbf{v}_i \triangleq \mathbf{u}_i - \mathbf{v}_i, \quad \delta\lambda_i \triangleq w_i - \lambda_i \ . \tag{26}$$

*Proof.* Note that from the definition of eigenvectors and eigenvalues, we have

$$\hat{\mathbf{C}}\mathbf{u}_i = w_i \mathbf{u}_i . \tag{27}$$

We can rewrite (27) in terms of perturbations with respect to the ensemble covariance matrix $\mathbf{C}$ and the outputs of its eigendecomposition as follows:

$$(\hat{\mathbf{C}} - \mathbf{C})(\mathbf{v}_i + \delta\mathbf{v}_i) + \mathbf{C}(\mathbf{v}_i + \delta\mathbf{v}_i) = (\lambda_i + \delta\lambda_i)(\mathbf{v}_i + \delta\mathbf{v}_i) , \tag{28}$$

where we have used $w_i = \lambda_i + \delta\lambda_i$ and $\mathbf{u}_i = \mathbf{v}_i + \delta\mathbf{v}_i$. Using the fact that $\mathbf{C}\mathbf{v}_i = \lambda_i\mathbf{v}_i$ and rearranging the terms in (28), we have

$$(\hat{\mathbf{C}} - \mathbf{C})\mathbf{v}_i = (\lambda_i\mathbf{I}_m - \mathbf{C})\delta\mathbf{v}_i + \delta\lambda_i(\mathbf{v}_i + \delta\mathbf{v}_i) - (\hat{\mathbf{C}} - \mathbf{C})\delta\mathbf{v}_i . \tag{29}$$

By setting $\mathbf{E} = \hat{\mathbf{C}} - \mathbf{C}$ and $\beta_i = \lambda_i\mathbf{I}_m - \mathbf{C}$, we can rewrite (29) as

$$\mathbf{E}\mathbf{v}_i = \beta_i\delta\mathbf{v}_i + \delta\lambda_i\mathbf{v}_i + (\delta\lambda_i\mathbf{I}_m - \mathbf{E})\delta\mathbf{v}_i . \tag{30}$$

$\square$

Next, we leverage Lemma 1 to complete the proof of Theorem 2.

## Proof of Theorem 2

*Proof.* To start with, we note that the coVariance filters with respect to $\hat{\mathbf{C}}$ and $\mathbf{C}$ are given by

$$\mathbf{H}(\hat{\mathbf{C}}) = \sum_{k=0}^{m} h_k \hat{\mathbf{C}}^k \quad \text{and} \quad \mathbf{H}(\mathbf{C}) = \sum_{k=0}^{m} h_k \mathbf{C}^k . \tag{31}$$

We aim to study the stability of the coVariance filters by analyzing the difference between $\mathbf{H}(\hat{\mathbf{C}})$ and $\mathbf{H}(\mathbf{C})$. For this purpose, we next establish the first order approximation for $\hat{\mathbf{C}}^k$ in terms of $\mathbf{C}$ and $\mathbf{E}$. Using $\hat{\mathbf{C}} = \mathbf{C} + \mathbf{E}$, the first order approximation of $\hat{\mathbf{C}}^k$ is given by

$$(\mathbf{C} + \mathbf{E})^k = \mathbf{C}^k + \sum_{r=0}^{m} \mathbf{C}^r\mathbf{E}\mathbf{C}^{k-r-1} + \tilde{\mathbf{E}} , \tag{32}$$

where $\tilde{\mathbf{E}}$ satisfies $\|\tilde{\mathbf{E}}\| \leq \sum_{r=2}^{k} \binom{k}{r}\|\mathbf{E}\|^r\|\mathbf{C}\|^{k-r}$. Using (32), we have

$$\mathbf{H}(\hat{\mathbf{C}}) - \mathbf{H}(\mathbf{C}) = \sum_{k=0}^{m} h_k[(\mathbf{C} + \mathbf{E})^k - \mathbf{C}^k] , \tag{33}$$

$$= \sum_{k=0}^{m} h_k \sum_{r=0}^{k-1} \mathbf{C}^r\mathbf{E}\mathbf{C}^{k-r-1} + \tilde{\mathbf{D}} , \tag{34}$$

where $\tilde{\mathbf{D}}$ satisfies $\|\tilde{\mathbf{D}}\|^2 = \mathcal{O}(\|\mathbf{E}\|^2)$ [19]. The focus of our subsequent analysis will be the first term in (34). For a random data sample $\mathbf{x} = [x_1, \ldots, x_m]^\mathsf{T}$, such that, $\|\mathbf{x}\| < R$, for some $R > 0$ and $\mathbf{x} \in \mathbb{R}^{m \times 1}$, its VFT with respect to $\mathbf{C}$ is given by $\tilde{\mathbf{x}} = \mathbf{V}^\mathsf{T}\mathbf{x}$, where $\tilde{\mathbf{x}} = [\tilde{x}_1, \ldots, \tilde{x}_m]^\mathsf{T}$. The relationship $\tilde{\mathbf{x}}$ and $\mathbf{x}$ can be expressed as

$$\mathbf{x} = \sum_{i=1}^{m} \tilde{x}_i\mathbf{v}_i . \tag{35}$$

Multiplying both sides in (34) by $\mathbf{x}$ and by leveraging (35), we get

$$[\mathbf{H}(\hat{\mathbf{C}}) - \mathbf{H}(\mathbf{C})]\mathbf{x} = \sum_{k=0}^{m} h_k \sum_{r=0}^{k-1} \mathbf{C}^r\mathbf{E}\mathbf{C}^{k-r-1}\mathbf{x} + \tilde{\mathbf{D}}\mathbf{x} , \tag{36}$$

$$= \sum_{i=1}^{m} \tilde{x}_i \sum_{k=0}^{m} h_k \sum_{r=0}^{k=1} \mathbf{C}^r\mathbf{E}\mathbf{C}^{k-r-1}\mathbf{v}_i + \tilde{\mathbf{D}}\mathbf{x} , \tag{37}$$

$$= \sum_{i=1}^{m} \tilde{x}_i \sum_{k=0}^{m} h_k \sum_{r=0}^{k-1} \mathbf{C}^r\lambda_i^{k-r-1}\mathbf{E}\mathbf{v}_i + \tilde{\mathbf{D}}\mathbf{x} , \tag{38}$$

where we have used $\mathbf{C}\mathbf{v}_i = \lambda_i \mathbf{v}_i$ in the transition from (37) to (38). We focus only on the first term in (38) and leverage the result from Lemma 1 that expands $\mathbf{E}\mathbf{v}_i$ to get

$$\sum_{i=1}^{m} \tilde{x}_i \sum_{k=0}^{m} h_k \sum_{r=0}^{k-1} \mathbf{C}^r \lambda_i^{k-r-1} \mathbf{E}\mathbf{v}_i = \underbrace{\sum_{i=1}^{m} \tilde{x}_i \sum_{k=0}^{m} h_k \sum_{r=0}^{k-1} \mathbf{C}^r \lambda_i^{k-r-1} \beta_i \delta \mathbf{v}_i}_{\text{Term 1}}$$

$$+ \underbrace{\sum_{i=1}^{m} \tilde{x}_i \sum_{k=0}^{m} h_k \sum_{r=0}^{k-1} \mathbf{C}^r \lambda_i^{k-r-1} \delta \lambda_i \mathbf{v}_i}_{\text{Term 2}}$$

$$+ \underbrace{\sum_{i=1}^{m} \tilde{x}_i \sum_{k=0}^{m} h_k \sum_{r=0}^{k-1} \mathbf{C}^r \lambda_i^{k-r-1} (\delta \lambda_i \mathbf{I}_m - \mathbf{E}) \delta \mathbf{v}_i}_{\text{Term 3}} . \qquad (39)$$

Next, we analyze term 1, term 2, and term 3 in (39) separately.

**Analysis of Term 1 in** (39): In the analysis of term 1, we start by noting that

$$\beta_i = \lambda_i \mathbf{I}_m - \mathbf{C} , \qquad (40)$$

$$= \sum_{j=1}^{m} (\lambda_i - \lambda_j) \mathbf{v}_j \mathbf{v}_j^\mathsf{T} , \qquad (41)$$

$$= \mathbf{V}(\lambda_i \mathbf{I}_m - \Lambda) \mathbf{V}^\mathsf{T} . \qquad (42)$$

Using (42) and $\delta \mathbf{v}_i = \mathbf{u}_i - \mathbf{v}_i$ in term 1 in (39), we have

$$\sum_{i=1}^{m} \tilde{x}_i \sum_{k=0}^{m} h_k \sum_{r=0}^{k-1} \mathbf{C}^r \lambda_i^{k-r-1} \mathbf{V}(\lambda_i \mathbf{I}_m - \Lambda) \mathbf{V}^\mathsf{T} (\mathbf{u}_i - \mathbf{v}_i) . \qquad (43)$$

Using $\mathbf{C}^r = \mathbf{V}\Lambda^r \mathbf{V}^\mathsf{T}$ in (43), term 1 in (39) is equivalent to

$$\sum_{i=1}^{m} \tilde{x}_i \sum_{k=0}^{m} h_k \sum_{r=0}^{k-1} \lambda_i^{k-r-1} \mathbf{V}\Lambda^r (\lambda_i \mathbf{I}_m - \Lambda) \mathbf{V}^\mathsf{T} (\mathbf{u}_i - \mathbf{v}_i) , \qquad (44)$$

$$= \sum_{i=1}^{m} \tilde{x}_i \mathbf{V}\mathbf{L}_i \mathbf{V}^\mathsf{T} (\mathbf{u}_i - \mathbf{v}_i) , \qquad (45)$$

where $\mathbf{L}_i$ is a diagonal matrix whose $j$-th element is given by

$$[\mathbf{L}_i]_j = \sum_{k=0}^{m} h_k \sum_{r=0}^{k-1} (\lambda_i - \lambda_j) \lambda_i^{k-r-1} \lambda_j^r , \qquad (46)$$

$$= \sum_{k=0}^{m} h_k (\lambda_i - \lambda_j) \frac{\lambda_i^k - \lambda_j^k}{\lambda_i - \lambda_j} , \qquad (47)$$

$$= \sum_{k=0}^{m} h_k \lambda_i^k - \sum_{k=0}^{m} h_k \lambda_j^k , \qquad (48)$$

$$= h(\lambda_i) - h(\lambda_j) , \qquad (49)$$

where $h(\lambda_i)$ is the frequency response of the coVariance filter and is defined in (8). Therefore, we have $\mathbf{L}_i = \text{diag}([h(\lambda_i) - h(\lambda_j)]_j)$. Next, in (45), we note that

$$\mathbf{V}^\mathsf{T} (\mathbf{u}_i - \mathbf{v}_i) = [\mathbf{v}_1^\mathsf{T} (\mathbf{u}_i - \mathbf{v}_i), \cdots , \mathbf{v}_m^\mathsf{T} (\mathbf{u}_i - \mathbf{v}_i)]^\mathsf{T} . \qquad (50)$$

Using (50) and (49) in (45) and $\mathbf{v}_j^\mathsf{T} \mathbf{v}_i = 0, \forall j \neq i$, we deduce that the term 1 in (39) is equivalent to

$$\sum_{i=1}^{m} \tilde{x}_i \mathbf{V}\mathbf{L}_i \mathbf{V}^\mathsf{T} (\mathbf{u}_i - \mathbf{v}_i) = \sum_{i=1}^{m} \tilde{x}_i \mathbf{V}\mathbf{J}_i , \qquad (51)$$

where the $j$-th element of $\mathbf{J}_i$ is given by

$$[\mathbf{J}_i]_j = \begin{cases} 0 , & \text{if } j = i , \\ (h(\lambda_i) - h(\lambda_j))\mathbf{v}_j^\mathsf{T}\mathbf{u}_i , & \text{otherwise} \end{cases} . \tag{52}$$

For the stability analysis, we are interested in the norm of term 1. Therefore, by noting the equivalence between the term 1 in (39) and (51), after taking the uniform norm, we have

$$\left\| \sum_{i=1}^m \tilde{x}_i \sum_{k=0}^m h_k \sum_{r=0}^{k-1} \mathbf{C}^r \lambda_i^{k-r-1} \beta_i \delta\mathbf{v}_i \right\| = \left\| \sum_{i=1}^m \tilde{x}_i \mathbf{V}\mathbf{J}_i \right\| , \tag{53}$$

$$\leq \sum_{i=1}^m |\tilde{x}_i| \max_{j,i\neq j} \|h(\lambda_i) - h(\lambda_j)\| \|\mathbf{v}_j^\mathsf{T}\mathbf{u}_i\| . \tag{54}$$

Note that $\mathbf{v}_j^\mathsf{T}\mathbf{u}_i$ is the inner product between the eigenvector $\mathbf{v}_j$ of ensemble covariance matrix $\mathbf{C}$ and the eigenvector $\mathbf{u}_i$ of the sample covariance matrix $\hat{\mathbf{C}}$. The bounds on $\mathbf{v}_j^\mathsf{T}\mathbf{u}_i$ in terms of the number of data samples $n$ have been studied in the existing literature. Here, we leverage the result from [30, Theorem 4.1] to conclude that if $\mathsf{sgn}(\lambda_j - \lambda_i)2w_j > \mathsf{sgn}(\lambda_j - \lambda_i)(\lambda_j - \lambda_i)$ for $\lambda_i \neq \lambda_j$, the condition

$$\left\| \sum_{i=1}^m \tilde{x}_i \sum_{k=0}^m h_k \sum_{r=0}^{k-1} \mathbf{C}^r \lambda_i^{k-r-1} \beta_i \delta\mathbf{v}_i \right\| \leq \sum_{i=1}^m |\tilde{x}_i| \max_{j,i\neq j} |h(\lambda_i) - h(\lambda_j)| \frac{2k_i}{n^{1/2-\varepsilon}|\lambda_i - \lambda_j|} , \tag{55}$$

is true with probability at least $\left(1 - \frac{1}{n^{2\varepsilon}}\right)$ for some $\varepsilon \in (0, 1/2]$, where $k_i = \left(\mathbb{E}[\|\mathbf{X}\mathbf{X}^\mathsf{T}\mathbf{v}_i\|_2^2] - \lambda_i^2\right)^{\frac{1}{2}}$. Furthermore, we note that the condition $\mathsf{sgn}(\lambda_j - \lambda_i)2w_j > \mathsf{sgn}(\lambda_j - \lambda_i)(\lambda_j - \lambda_i)$ is satisfied with probability at least $1 - \frac{2k_i^2}{|\lambda_i - \lambda_j|}$ [30, Corollary 4.2], which via a union bound and first order approximation from Taylor series implies that (55) is true with probability at least $1 - \frac{1}{n^{2\varepsilon}} - \frac{2\kappa m}{n}$ for $\kappa$ defined in (17). Therefore, for a coVariance filter with the property

$$\max_{i,j\in\{1,\ldots,m\},i\neq j} \frac{|h(\lambda_i) - h(\lambda_j)|}{|\lambda_i - \lambda_j|} \leq \frac{M}{k_i} , \tag{56}$$

for some real constant $M > 0$, the condition in (55) is equivalent to

$$\left\| \sum_{i=1}^m \tilde{x}_i \sum_{k=0}^m h_k \sum_{r=0}^{k-1} \mathbf{C}^r \lambda_i^{k-r-1} \beta_i \delta\mathbf{v}_i \right\| \leq \frac{2M}{n^{\frac{1}{2}-\varepsilon}} \sum_{i=1}^m |\tilde{x}_i| , \tag{57}$$

which holds with probability at least $1 - \frac{1}{n^{2\varepsilon}} - \frac{2\kappa m}{n}$. Furthermore, note that $\sum_{i=1}^m |\tilde{x}_i| \leq \sqrt{m}\|\mathbf{x}\|_2$. If the random sample $\mathbf{x}$ satisfies $\|\mathbf{x}\|_2 \leq R$, then we have

$$\mathbb{P}\left( \left\| \sum_{i=1}^m \tilde{x}_i \sum_{k=0}^m h_k \sum_{r=0}^{k-1} \mathbf{C}^r \lambda_i^{k-r-1} \beta_i \delta\mathbf{v}_i \right\| \leq \frac{2}{n^{\frac{1}{2}-\varepsilon}}\sqrt{m}MR \right) \geq 1 - \frac{1}{n^{2\varepsilon}} - \frac{2\kappa m}{n} , \tag{58}$$

for any $\varepsilon \in (0, 1/2]$.

**Analysis of Term 2 in** (39): Using $\mathbf{C}\mathbf{v}_i = \lambda_i\mathbf{v}_i$, we note that term 2 in (39) is equivalent to

$$\sum_{i=1}^m \tilde{x}_i \sum_{k=0}^m h_k \sum_{r=0}^{k-1} \mathbf{C}^r \lambda_i^{k-r-1} \delta\lambda_i\mathbf{v}_i = \sum_{i=1}^m \tilde{x}_i \sum_{k=0}^m h_k \sum_{r=0}^{k-1} \lambda_i^{k-1} \delta\lambda_i\mathbf{v}_i , \tag{59}$$

$$= \sum_{i=1}^m \tilde{x}_i \sum_{k=0}^m k h_k \lambda_i^{k-1} \delta\lambda_i\mathbf{v}_i , \tag{60}$$

$$= \sum_{i=1}^m \tilde{x}_i h'(\lambda_i)\delta\lambda_i\mathbf{v}_i . \tag{61}$$

Next, using Weyl's theorem [38, Theorem 8.1.6], we note that $\|\mathbf{E}\| \leq \alpha$ implies that $|\delta\lambda_i| \leq \alpha$ for any $\alpha > 0$. For a random instance $\mathbf{x}$ of random vector $\mathbf{X}$ whose probability distribution is supported within a bounded region w.l.o.g, such that, $\|\mathbf{x}\| \leq 1$, we have

$$\mathbb{P}\left(\mathbf{E} \leq B\left(\frac{\|\mathbf{C}\|\sqrt{\log m + u}}{\sqrt{n}} + \frac{(1 + \|\mathbf{C}\|)(\log m + u)}{n}\right)\right) \geq 1 - 2^{-u},\tag{62}$$

for some constant $B > 0$ and $u > 0$. The result in (62) follows directly from [39, Theorem 5.6.1]. Therefore, using (61), we have

$$\left\|\sum_{i=1}^{m} \tilde{x}_i \sum_{k=0}^{m} h_k \sum_{r=0}^{k-1} \mathbf{C}^r \lambda_i^{k-r-1} \delta\lambda_i \mathbf{v}_i\right\| \leq \sum_{i=1}^{m} |\tilde{x}_i||h'(\lambda_i)||\delta\lambda_i|\|\mathbf{v}_i\|.\tag{63}$$

Using (62), $|h'(\lambda_i)| \leq M/k_{\mathsf{min}}$ (where $k_{\mathsf{min}} = \min_{i \in \{1,\ldots,m\}, \lambda_i > 0} k_i$) from (56), and $\|\mathbf{v}_i\| = 1$, we have

$$\mathbb{P}\left(\left\|\sum_{i=1}^{m} \tilde{x}_i \sum_{k=0}^{m} h_k \sum_{r=0}^{k-1} \mathbf{C}^r \lambda_i^{k-r-1} \delta\lambda_i \mathbf{v}_i\right\|\right.$$
$$\left.\leq \frac{A}{k_{\mathsf{min}}}\sqrt{m}M\left(\frac{\|\mathbf{C}\|\sqrt{\log m + u}}{\sqrt{n}} + \frac{(1 + \|\mathbf{C}\|)(\log m + u)}{n}\right)\right) \geq 1 - 2^{-u},\tag{64}$$

for some constant $A > 0$ and $u > 0$.

**Analysis of Term 3 in** (39): We remark that the term 3 in (39) consists of second order error terms that diminish faster with the number of samples $n$ as compared to term 1 and term 2. To illustrate this, we note two facts. First, from (62) and Weyl's theorem, we note that $\|\delta\lambda_i\mathbf{I}_m - \mathbf{E}\| \leq 2\|\mathbf{E}\|$ and $\|\mathbf{E}\| \simeq \mathcal{O}(1/\sqrt{n})$ with high probability. Secondly, from the existing literature [40, 41], we note that if $\hat{\mathbf{C}}$ follows the Wishart distribution, we have

$$\mathbb{E}[\|\delta\mathbf{v}_i\|] = 0\tag{65}$$

and

$$\mathbb{E}[n\|\delta\mathbf{v}_i\|^2] = \sum_{j \neq i} \frac{\lambda_i\lambda_j}{(\lambda_i - \lambda_j)^2}.\tag{66}$$

Let $\sigma_{\mathbf{v}_i}^2 \triangleq \sum_{j \neq i} \frac{\lambda_i\lambda_j}{(\lambda_i - \lambda_j)^2}$. Using Chebyshev's inequality, we note that

$$\mathbb{P}\left(\sqrt{n}\|\delta\mathbf{v}_i\| \geq \gamma\sigma_{\mathbf{v}_i}\right) \leq \frac{1}{\gamma^2},\tag{67}$$

for any constant $\gamma > 1$. Furthermore, (67) is equivalent to

$$\mathbb{P}\left(\|\delta\mathbf{v}_i\| \leq \frac{\gamma}{\sqrt{n}}\sigma_{\mathbf{v}_i}\right) \geq 1 - \frac{1}{\gamma^2},\tag{68}$$

which implies that $\|\delta\mathbf{v}_i\| = \mathcal{O}(1/\sqrt{n})$ with high probability. Therefore, the second order error term $(\delta\lambda_i\mathbf{I}_m - \mathbf{E})\delta\mathbf{v}_i$ scales as $\mathcal{O}(1/n)$, which diminishes faster with $n$ as compared to terms 1 and 2, that individually scale as $\mathcal{O}(1/n^{1/2-\varepsilon})$ for $\varepsilon \in (0, 1/2]$ and $\mathcal{O}(1/\sqrt{n})$, respectively.

The proof of Theorem 2 is completed by noting that the condition on $\|[\mathbf{H}(\hat{\mathbf{C}}) - \mathbf{H}(\mathbf{C})]\mathbf{x}\|$ reduces to the condition on operator norm $\|[\mathbf{H}(\hat{\mathbf{C}}) - \mathbf{H}(\mathbf{C})]\|$ for any $\|\mathbf{x}\| \leq 1$ and that the terms scaling at $1/\sqrt{n}$ or slower in (58) and (64) dominate the scaling behavior of the upper bound on $\|[\mathbf{H}(\hat{\mathbf{C}}) - \mathbf{H}(\mathbf{C})]\|$. $\qquad\square$

# D  VNN architecture

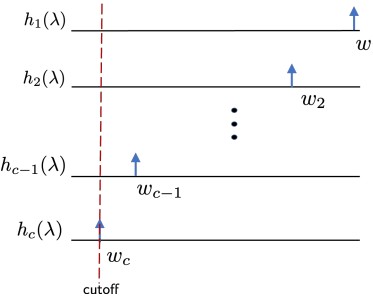

Figure 3: Filter response for different coVariance filters sufficient to implement PCA transformation that includes $c$ largest eigenvalues of the covariance matrix $\hat{\mathbf{C}}_n$ according to Theorem 1.

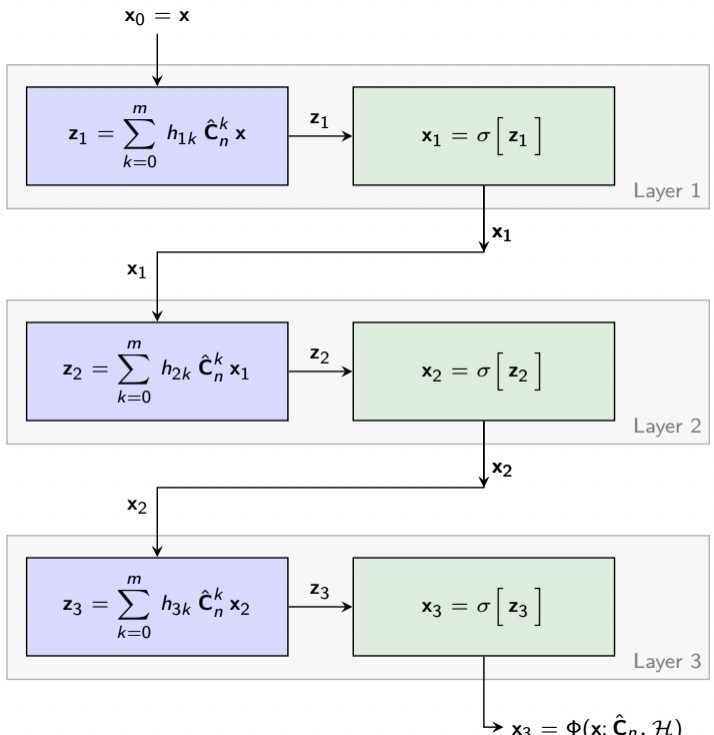

Figure 4: A 3-layer VNN architecture.

# E  Additional Experiments and Discussions

In this section, we provide additional discussions on the age prediction task using cortical thickness and related experiments on the synthetic data. In Section E.1, we briefly discuss the motivation behind studying age prediction and PCA-based statistical analysis in this context. In Section E.2, we provide additional details on cortical thickness data acquisition. In Section E.3, we report the results for stability analysis of VNNs and PCA-regression models for FTDC100 ($m = 100$) and FTDC300 ($m = 300$) datasets. In Section E.4, we study the stability of VNNs on two simulated settings that include non-linear and linear data models. In Section E.5, we include additional figures that supplement the VNN transferability results in Fig. 2.

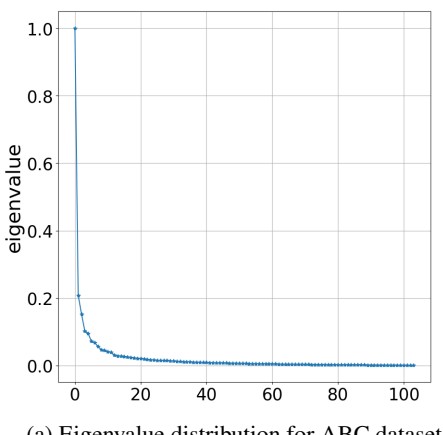
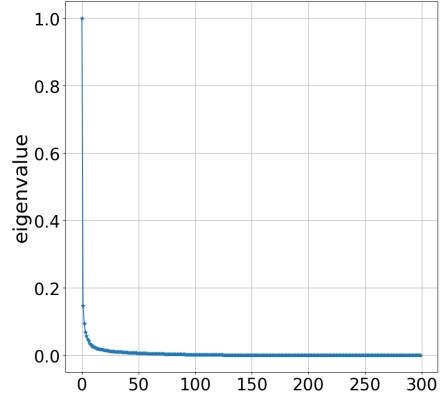

(a) Eigenvalue distribution for ABC dataset.     (b) Eigenvalue distribution for FTDC300 dataset.

Figure 5: Eigenvalue distributions.

## E.1 Predicting Brain Age with Cortical Thickness Data

Brain age estimation using various modalities of magnetic resonance imaging (MRI) is an active area of research [42, 43, 44]. A promising application of brain age prediction is early detection of neurodegenerative diseases (such as Alzheimer's, Huntingson's disease) which may manifest themselves as error in age prediction in pathological contexts by machine learning models trained on healthy subjects. Cortex anatomical measures extracted from structural MRI scans have shown promising results in age prediction in existing studies [43, 44]. Cortical measures across the brain usually have high collinearity and therefore, the age prediction pipeline consisting of dimension reduction on cortical features and regression models is commonly employed. The utility of dimension reduction is apparent in ABC and FTDC datasets used in this paper as well from Fig. 5a and Fig. 5b, where we observe that the eigenvalue distribution for ABC and FTDC300 datasets is skewed and therefore, PCA based dimensionality reduction is well-motivated.

The studies in [43] and [44] report age prediction results on different datasets using various approaches, such as PCA-based regression (similar to the PCA-regression models evaluated in this paper), ridge regression, lasso based regression, and fully connected neural networks. All these approaches show comparable performances in predicting age using cortical anatomical measures [43]. The objective of our experiments is to demonstrate the properties of VNNs, such as stability and transferability, against other covariance matrix driven statistical analyses. In this context, PCA-based regression models are the natural baselines for comparison against VNNs.

## E.2 Cortical Thickness Data Acquisition

Cortical thickness measures in regions of interest (ROIs) were derived using $0.8 - 1$ mm isotropic T1-weighted MRI. The complete pipeline for cortical thickness extraction for ABC data is similar to that in [45].

## E.3 Stability of VNNs on FTDC100 and FTDC300 datasets

In this subsection, we study the stability of VNNs for FTDC100 and FTDC300 datasets in a similar fashion as for ABC dataset in Section 3. We split each dataset into a $90/10$ train/test split, such that, we have $153$ samples in the training set and $17$ samples in the test set. The nominal VNN and PCA-regression models for FTDC100 and FTDC300 are trained on their respective sample covariance matrices $\hat{\mathbf{C}}_{153}$ derived from cortical thickness data. The architecture and hyperparamaters for VNN training for FTDC100 and FTDC300 are same as that reported in Section 5.2. Figure 6 a) and b) show the variance in MAE over training and test sets for nominal models based on VNN and PCA-regression with respect to $\hat{\mathbf{C}}_{153}$ for FTDC100 dataset. Figure 6 c) and d) show the variance in MAE over training and test sets for nominal models based on VNN and PCA-regression with respect to $\hat{\mathbf{C}}_{153}$ for FTDC300 dataset. For both datasets, we observe that VNNs are stable with respect to the perturbations in the sample covariance matrix. Also, note that our theoretical results in Theorem 5.1

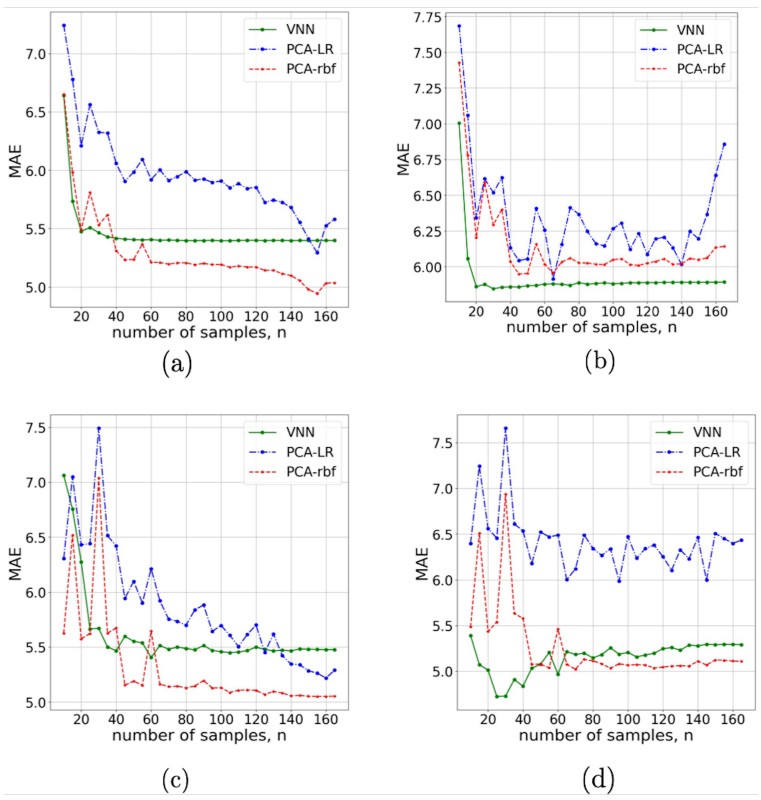

Figure 6: Stability on FTDC100 ((a) for training set and (b) for test set) and FTDC300 ((c) for training set and (d) for test set).

and Theorem 3 suggest stability of coVariance filters and VNNs for $n > m$. Our observations from the experiments on FTDC300 dataset show evidence that VNNs retain stability even when this condition is violated. Significant randomness is induced into the performance of $\mathsf{PCA-LR}$ when the principal components are perturbed due to perturbations in $\hat{\mathbf{C}}_{153}$ for both datasets. However, in Fig. 6, we also observe that $\mathsf{PCA\text{-}rbf}$ retains stability in performance when $\hat{\mathbf{C}}_{153}$ is replaced by $\hat{\mathbf{C}}_{n'}$ for $n' \in [65, 152]$.

### E.4 Stability of VNNs on Synthetic Data

We consider two settings for synthetic data.

**Friedman regression problem**: This setting is described as 'Friedman1' in [46] and is generated using the routine `sklearn.datasets.make_friedman1` in python. In the simulated dataset, we have $m$ independent features or predictors, each sampled uniformly from the range $[0, 1]$. Out of $m$ predictors, any 5 are used to generate the response variable and others are independent of the response. If the 5 relevant predictors are given by $x_1, x_2, x_3, x_4$, and $x_5$, they are related to the response $y$ as

$$y = 10\sin(\pi x_1 x_2) + 20(x_3 - 0.5)^2 + 10x_4 + 5x_5 + \vartheta , \tag{69}$$

where $\vartheta$ represents noise. We generate a dataset of $n = 1000$ samples using $m = 100$ features and noise distributed according to $\mathcal{N}(0, 1)$. Figure 7 a) shows the distribution of eigenvalues in the covariance matrix for the dataset used. The dataset is split into a $90/10$ train/test split and a sample covariance matrix $\hat{\mathbf{C}}_{900}$ is evaluated from the features in the training set. Next, we perform regression against the response using VNN, $\mathsf{PCA\text{-}LR}$, and $\mathsf{PCA\text{-}rbf}$ models. For VNNs, we use the same architecture and hyperparameters as described in Section 5.1. Using different permutations of the training set, we train 100 nominal models for VNN, $\mathsf{PCA\text{-}LR}$, and $\mathsf{PCA\text{-}rbf}$.

Figure 7 b) plots the variation in mean performance of the nominal models with respect to perturbations in the sample covariance matrix. The last data point in Fig. 7 b) corresponds to the performance

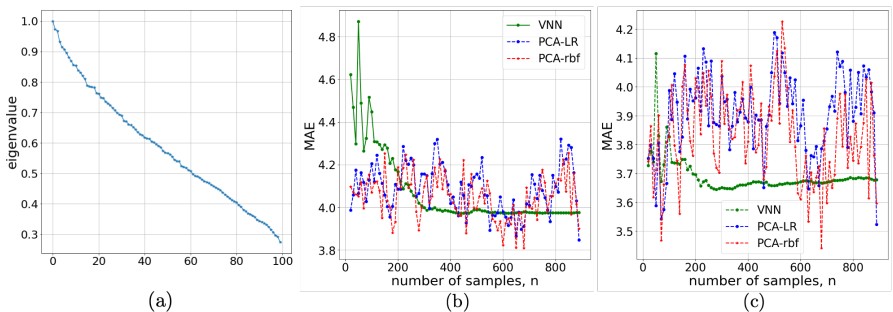

Figure 7: Friedman regression problem.

of nominal models trained on $\hat{\mathbf{C}}_{900}$. When $\hat{\mathbf{C}}_{900}$ is replaced with $\hat{\mathbf{C}}_{n'}$ for $n' \in [5, 899]$, we observe that the VNN performance retains stability for sample covariance matrices generated with $n' > 300$ samples. However, considerable randomness is introduced in the performance of PCA-regression models when the principal components are evaluated from $\hat{\mathbf{C}}_{n'}$ for $n' \neq 900$. Similar phenomenon is observed for the performance on the test set in Fig. 7 c).

**Linear regression problem**: In this setting, we generate random linear regression problems using the routine `sklearn.datasets.make_regression` in python, which allows us to specify the number of informative features; effective rank of the input dataset, i.e., the approximate number of singular vectors required to explain most of the input data by linear combinations; tail parameter which is the relative importance of the fat noisy tail of the singular values profile; and noise. In our experiments, we set the input dimension $m = 100$, the dimension of the response to be 1, number of samples $n = 1000$, number of informative features to be 20, effective rank of the input dataset to be 25 and noise to be distributed as $\mathcal{N}(0, 3)$. Since the stability of VNNs depends on the eigengap and kurtosis of underlying distribution of the data in direction of eigenvectors according to Theorem 3, we also aim to study the effects on VNN distribution with respect to variation in the strength of the tail of the eigenvalue distribution. To this end, we generate two datasets for linear regression with tail parameters set to 0.7 and 0.2. The eigenvalues of the covariance matrix for tail set to 0.7 are more spread out as compared to that for tail set to 0.2 (see Fig. 8 a) and Fig. 9 a)).

To evaluate the stability of VNNs, we split each dataset into a $90/10$ train/test split and generate sample covariance matrices $\hat{\mathbf{C}}_{900}$ for both. Next, we perform regression using VNN and PCA-LR models. For VNNs, we use the same architecture and hyperparameters as described in Section 5.1. Using different permutations of the training set, we obtain 100 nominal models for VNN and PCA − IR. Figure 8 b) plots the variations in mean MAE performances of the nominal models with respect to perturbations in the sample covariance matrix, with the last data point corresponding to $n = 900$, i.e., $\hat{\mathbf{C}}_{900}$. When $\hat{\mathbf{C}}_{900}$ is replaces with $\hat{\mathbf{C}}_{n'}$ for any $n' \in [5, 899]$, our experiments show that VNN performance is stable but significant randomness is induced into the performance of PCA-LR model. Similar phenomenon is observed for the performance on the test set in Fig. 8 c). Same discussion also follows for the results in Fig. 9 b)-c), where we have set the tail parameter for eigenvalue distribution to be 0.2. Comparison of MAE performances in Fig. 8 c) and Fig. 9 c) reveals that both VNN and PCA − LR perform significantly better for dataset with tail set to 0.2 than that for 0.7. This observation is along the expected lines as the first 25 eigenvalues and eigenvectors that contain the most information are more separated (and hence, more accurately estimated by the sample covariance matrix) for the dataset with tail parameter 0.2 as compared to the dataset with tail parameter 0.7. Moreover, VNNs retain the stability property for both datasets.

Finally, we also note that the observations above also extend to data with higher dimensions. Figure 10 illustrates the results on a setting for linear regression with $m = 1000$, number of informative features to be 20, effective rank of the input dataset to be 25 and noise to be distributed as $\mathcal{N}(0, 3)$. The tail is set to 0.7. The number of samples is 5000 of which 4500 are used to form the covariance matrix $\hat{\mathbf{C}}_{4500}$ for training PCA-regression and VNN models. Figure 10 a) illustrates the eigenvalue distribution for this setting. Figure 10 b) and c) show the results of stability analysis by perturbing $\hat{\mathbf{C}}_{4500}$ for training set and test set, respectively.

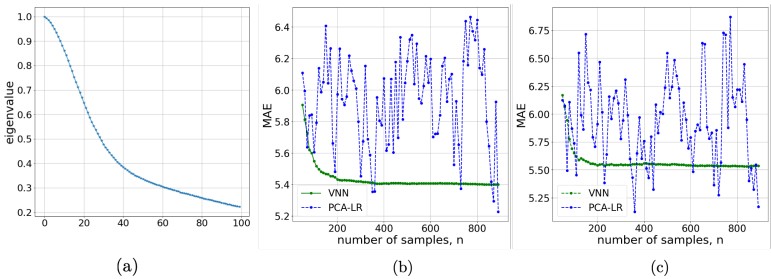

Figure 8: Stability of VNNs on linear regression problem (tail = 0.7).

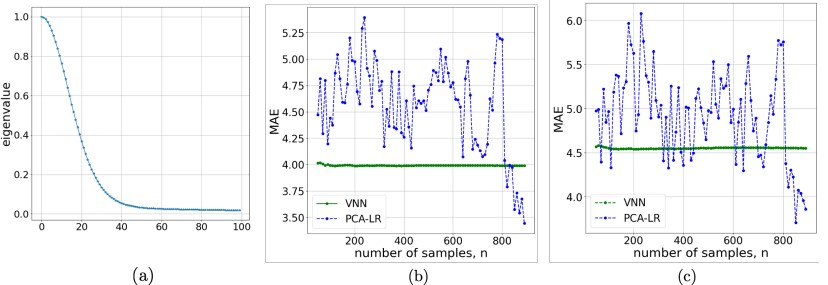

Figure 9: Stability of VNNs on linear regression problem (tail = 0.2).

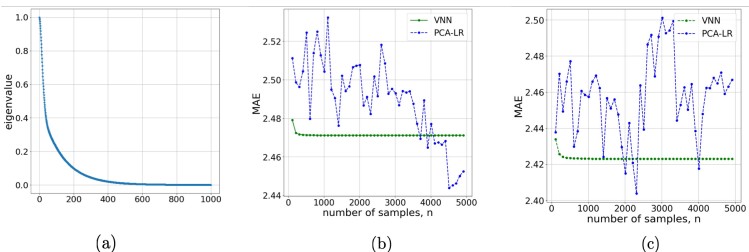

Figure 10: Stability of VNNs on linear regression problem with $m = 1000$ (tail = 0.7).

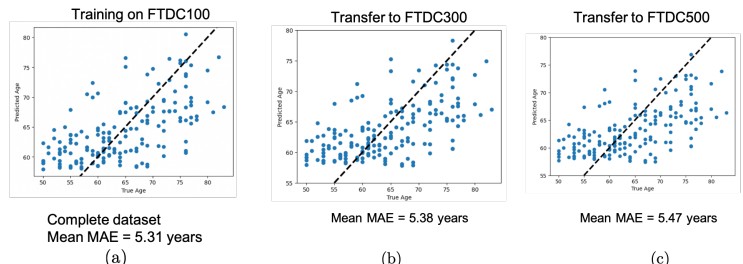

Figure 11: Transferability of VNN trained on FTDC100 to FTDC300 and FTDC500 datasets.

### E.5 Transferability of VNNs

In Fig. 11, we show the predicted age vs true age plots corresponding to the first rows of the matrices in Fig. 2, i.e., when the VNN is trained on FTDC100 dataset and its transferability is evaluated on FTDC300 and FTDC500 datasets. Similar plots are observed for transferability in other reported settings as well.