# OpenReview forum: "coVariance Neural Networks"
_NeurIPS.cc/2022/Conference — NeurIPS 2022 Accept_

### Official Review · Reviewer_NdkQ · 2022-07-11

**Rating:** 3
**Confidence:** 4
**Soundness:** 2 fair
**Presentation:** 2 fair
**Contribution:** 2 fair

**Summary:**

The authors propose a coVariance Neural Network (VNN) in analogy to graph neural networks based on covariance matrices of datasets, similar to what is used in PCA. The authors also introduce a “covariance transform” which projects a new datapoint into the PCA space by projecting to the eigenvectors of the covariance operator $UX$ and a covariance filter which learns coefficients of polynomial of the covariance matrix.  They show that using covariance filters one can recover the covariance transform. They also show that this polynomial filter is stable with respect to perturbations in the sampled data from which the sampled covariance matrix is taken.

**Questions:**

What are other applications of the covariance filter that would not be subsumed by SVD low rank approximations?

**Limitations:**

I do not see a discussion of limitations in this manuscript.

**Strengths And Weaknesses:**

Strengths:

The authors correctly note the similarity between the covariance matrix and an adjacency matrix of a graph, i.e., positive semidefinite matrices whose eigenvectors and values have specific properties in terms of describing the heterogeneity in the data. In fact it can be argued the other way that graph fourier transforms and graph signal processing are the result of a “kernel trick” based on PCA which is far older.  So in this sense GNNs/GCNs are a more general class of networks that subsumes VNNs.

The note that this results in a stable PCA is interesting although over the years there have been robust forms of PCA that have been developed including variations of Principal component pursuit.

Weakness:

However,  I do not believe their proposal constitutes a new neural network of any sort. I believe this is just an application of GCNs to the graph consisting of data features as nodes and data points as signals on the nodes (rather than vice versa). While this may be desirable for some types of analysis (see Tong et al. IEEE ICCASP 2022 for analysis of cells as signals over gene graphs), I do not believe it has advantages over robust PCA for this specific application.

The paper should potentially be reconfigured to simply talk about an application of GCNs/GNNs to feature covariance matrices and situations where that could be useful.

Further a key weakness here is regarding relationships between data features as defined by covariances. This is a strictly linear relationship, if this was changed to mutual information or some other relationship type then indeed a more complex relational graph would be necessary and this is precisely where GNNs/GCNs have contributed.

Other uses of the covariance filter seem to amount to low-rank approximations done via SVD and again I don't see much advantage in using a neural network for this kind of linear operation.

---

> ### Author Response · Authors · 2022-07-30
> **Response to Reviewer NdkQ (Part 1: Robust PCA and VNN)**
>
> We thank the reviewer for their insightful feedback. We also find the connections made by the reviewer between kernel PCA and covariance Fourier transform very interesting. We discuss the comparison between robust PCA and VNNs in this comment.  Other concerns on motivation, relevance of covariance/correlation matrices and comparison with linear SVD are addressed separately in Part 2.
>
> 1. **Robust PCA and VNNs have different objectives:** Robust PCA and the method in our paper tackle fundamentally distinct problems and therefore, the notions of stability or robustness in these approaches are different. Specifically, the notion of stability in our work pertains to outputs of VNNs being robust to the statistical uncertainty in the covariance matrix estimation due to finite sample size. In contrast, robust PCA using principal components pursuit aims to recover low rank structure in a given, high-dimensional data when the data is corrupted by gross errors or outliers.  We further elaborate on the differences between robust PCA and VNNs further by discussing the settings in robust PCA literature from a few seminal works in [a,b,c] and VNNs separately.
>
>    **_Robust PCA_**:  Let’s assume that the data matrix is given by $X$. Robust PCA is typically studied when $X$ conforms to the decomposition $X = L_0+S_0+Z_0$, where $L_0$  is the low rank structure in the data $X$, $S_0$ is a matrix that models _gross_ outliers (e.g. due to missing data, adversarial behavior, defects in in data collection), and $Z_0$ is random noise [b,c]. The robust PCA framework using principal component pursuit aims to find an estimate $L$ for $L_0$ by solving an optimization problem that is guaranteed to recover $L_0$ perfectly under assumptions on the sparsity of singular vectors or principal components of $L_0$ and the structure of $S_0$ (for instance, $S_0$ is not desirable to be low rank in [a,b]). In effect, robust PCA aims to recover the PCA decomposition for $X - S_0$ while being stable to noise $Z_0$ [b,c].
>
>
>    **_VNN_**: In our paper, we discuss potential instability of PCA-based statistical models due to perturbations in the data, for e.g., by adding a new sample to the dataset. However, we note that in contrast to robust PCA, the perturbations in the data are not driven by corrupted data points. Furthermore, ill-defined eigenvalues and eigenvectors of the sample covariance matrix can be the source of instability in statistical inference in this scenario.
>
>    Our theoretical contribution can be summarised as follows: _given a sample covariance matrix $\hat C_n$ of data $X$, we have a VNN output given by $\Phi(X; \hat C_n, {\cal H})$, where ${\cal H}$ is the set of covariance filters learnt for the inference task. Through our analysis, we establish that if $\hat C_n$ is replaced with another sample covariance matrix $\hat C_m$ (estimated from a different data matrix with the same underlying distribution as $X$), the VNN output will be stable, i.e., the difference between $\Phi(X; \hat C_n, {\cal H})$ and $\Phi(X; \hat C_m, {\cal H})$ will be bounded._
>
>    To establish this result, we leverage the perturbation theory of sample covariance matrices and identify that the closeness of eigenvalues of the covariance matrix and kurtosis of the underlying distribution for data matrix $X$ determine the design of filters ${\cal H}$ that ensures the stability of VNN outputs. Therefore, stability is an inherent property of VNNs and unlike robust PCA, we do not perform or leverage any decomposition or denoising in the training procedure. We have summarised the above differences between robust PCA and VNN in the Related Work section in Appendix E in the revised manuscript.
>
>
>    _[a] Candès, Emmanuel J., Xiaodong Li, Yi Ma, and John Wright. "Robust principal component analysis?." Journal of the ACM (JACM) 58, no. 3 (2011): 1-37._
>
>    _[b] Zhou, Z., Li, X., Wright, J., Candes, E., & Ma, Y. (2010, June). Stable principal component pursuit. In 2010 IEEE international symposium on information theory (pp. 1518-1522). IEEE._
>
>    _[c] Xu, Huan, Constantine Caramanis, and Sujay Sanghavi. "Robust PCA via outlier pursuit." Advances in neural information processing systems 23 (2010)._
>
>
> 2. **Robust PCA is not transferable:** Furthermore, there is no notion of ‘transferability’ in robust PCA, i.e., there is no possibility of generalising the framework to datasets of different dimensions. Our experiments on multi-resolution datasets in Section 5.2 show that VNNs learnt on a dataset with dimension 100 can be transferred to a higher resolution dataset of 500 without any retraining and vice-versa. We have also clarified this in Section 5.2.

---

> ### Author Response · Authors · 2022-07-30
> **Response to Reviewer NdkQ (Part 2)**
>
> In this comment, we continue our response to address the following concerns:
>
> 3. **Motivation behind studying VNN separately from GCNs:**
> We agree that VNNs are indeed implemented as GCNs with covariance matrices as graphs, where the data features act as nodes and data points act as signals on the nodes. Due to the ubiquity of PCA in statistical analysis and widespread use of covariance matrices to model relationships in various domains, we believe that the link between VNNs and PCA is a significant observation that merits the study of VNNs independently of GCNs. We also remark that independent study of related inference approaches to bring into focus the significant concepts or domain-specific novelties have precedence in both machine learning and statistics. For instance, graph neural networks and convolutional neural networks (CNN) are commonly studied independently even as the graph convolutions supersede convolution operations in CNN and the images on which CNNs primarily operate can be thought of as a grid graph. Furthermore, application-specific variations of PCA are studied independently, such as Karhunen–Loève transform in signal processing [d] and empirical orthogonal functions in atmospheric science [e]. We have clarified the relationship between VNN and GCN in the related work in Appendix E in the revised manuscript, which will be included in the introduction as Section 1.3 in the final version of the paper.
>
>     _[d] Dony, R. "Karhunen-loeve transform." The transform and data compression handbook 1.1-34 (2001): 29._
>
>     _[e] Hannachi, Abdel, Ian T. Jolliffe, and David B. Stephenson. "Empirical orthogonal functions and related techniques in atmospheric science: A review." International Journal of Climatology: A Journal of the Royal Meteorological Society 27.9 (2007): 1119-1152._
> 4. **VNNs and linear SVD approximation:**
> VNNs as neural networks are capable of inference tasks with non-linearity. This is because every layer of VNN consists of a covariance filter  and a pointwise nonlinearity function (e.g. ReLU). Therefore, VNNs subsume any inference tasks that use linear SVD approximation as the preliminary step.
>
>    We also re-iterate that traditional feature selection methods like _PCA and SVD are not transferable_, i.e., if the dimension of the dataset changes, PCA and SVD need to be performed again and cannot leverage the features extracted on a dataset of a different dimension. The transferability property of VNNs is inherited from GCNs and we illustrate it in our experiments in Section 5.2.
>
> 5. **Relevance of Correlation/Covariance Matrices:**
> It is true that covariance matrices cover only linear relationships between data features. However, we argue that our results regarding stability and transferability of VNNs in this context will be of interest to a wider audience that relies on PCA and correlation matrices for data analysis. Correlation or covariance matrices are very commonly used as graphs to model the brain connectivity and as inputs to graph neural networks when applied in neuroimaging applications and bioinformatics. This motivated our experiments in Section 5. Besides this application, correlation matrices are used for analyses in diverse fields such as traffic forecasting [f], environment monitoring [g], and natural language processing [h], to name a few.
>
>
>      _[f] Mallick, Tanwi, et al. "Dynamic graph neural network for traffic forecasting in wide area networks." 2020 IEEE International Conference on Big Data (Big Data). IEEE, 2020._
>
>       _[g] Cotta, Higor Henrique Aranda, Valdério Anselmo Reisen, and Pascal Bondon. "Identification of redundant air quality monitoring stations using robust principal component analysis." Environmental Modeling & Assessment 25.4 (2020): 521-530._
>
>      _[h] Malekzadeh, Masoud, et al. "Review of graph neural network in text classification." 2021 IEEE 12th Annual Ubiquitous Computing, Electronics & Mobile Communication Conference (UEMCON). IEEE, 2021._
>
> 6. **Computational complexity as potential limitation:** The computational cost for a covariance perceptron defined in (14) is given by $O(m^2 T F_{\sf in} F_{\sf out})$, where $T$ is the maximum number of filter taps in any filter in its associated filter bank and $m$ is the size of covariance matrix. Therefore, scalability to large covariance matrices is the most challenging aspect due to increased computational complexity in terms of $m$. We have added a brief discussion on the scalability of VNNs to large graphs as a potential limitation in the final version of the paper (see Remark 2 in the revised manuscript).
>
>
> We hope that we addressed the reviewer's concerns sufficiently, in which case, we would be grateful if your rating of our paper could be re-evaluated. We would be happy to clarify any additional concerns.

---

> > ### Comment · Reviewer_NdkQ · 2022-08-08
> > **New dimensions, new points.**
> >
> > The analogy of VNNs to GNNs with CNN vs GNN does not quite hold since graph convolutions are SIGNIFICANTLY different from image convolutions.
> >
> > If the dimension of the dataset changes, I believe that VNNs could incorporate this to a limited extent in the sense that the new dimension would be connected to a limited number of  original dimensions via the addition of the node in the graph. But this would not accurately re-evaluate the dependencies on other nodes without retraining.  Note that PCA is already "transferrable" to new datapoints in a way that most non-linear dimensionality reduction methods are not.
> >
> > If it admits non-linearity then this has to be related to non-linear dimensionality reduction which is not taken up in this paper.

---

> > > ### Author Response · Authors · 2022-08-08
> > > **Response: 'New dimensions, new points'**
> > >
> > > Thank you for considering our previous response. We address further concerns raised by the reviewer as follows.
> > >
> > > >*The analogy of VNNs to GNNs with CNN vs GNN does not quite hold since graph convolutions are SIGNIFICANTLY different from image convolutions.*
> > >
> > > Please note that there exists a rich literature in signal processing that provides a unified view for convolutional operations, where the convolutions over time (1-D or 1-Dimensional), space (2-D), and graphs are **specific instances of the same mathematical object** that exploits the symmetries and relationships in different data domains [i]. For instance, in seminal works on graph signal processing, the motivation for graph convolutions comes from rewriting time convolutions as graph convolutions on a directed line graph.
> > >
> > > Indeed, a number of works on graph signal processing and graph neural networks discuss the bridge between the graph convolutional operations and 2D convolutions (pertinent to CNN); see Section IV-C in [ii], Section 2.2 in [iii], Fig. 1 and Section V-B in [iv]. Moreover, in natural images represented as graphs, the covariance kernels recover the classical convolutional operations over images [v].
> > >
> > > In summary, we find it indisputable that the graph convolutions are generalisations of 2-D image convolutions or equivalently, 2-D convolutions are special cases of graph convolutions. **If we have misunderstood your statement, we would greatly appreciate further clarifications.**
> > >
> > >
> > > *i. Puschel, Markus, and José MF Moura. "Algebraic signal processing theory: Foundation and 1-D time." IEEE Transactions on Signal Processing 56.8 (2008): 3572-3585.*
> > >
> > > *ii. Ortega, A. et al. (2018). Graph signal processing: Overview, challenges, and applications. Proceedings of the IEEE, 106(5), 808-828.*
> > >
> > > *iii. Narang, Sunil K. et al. "Graph-wavelet filterbanks for edge-aware image processing." 2012 IEEE Statistical Signal Processing Workshop (SSP). IEEE, 2012.*
> > >
> > > *iv. Wu, Zonghan, et al. "A comprehensive survey on graph neural networks." IEEE transactions on neural networks and learning systems 32.1 (2020): 4-24.*
> > >
> > > *v. Bronstein, M. M. et al. (2017). Geometric deep learning: going beyond euclidean data. IEEE Signal Processing Magazine, 34(4), 18-42.*
> > >
> > > >*If the dimension of the dataset changes, I believe that VNNs could incorporate this to a limited extent in the sense that the new dimension would be connected to a limited number of original dimensions via the addition of the node in the graph. But this would not accurately re-evaluate the dependencies on other nodes without retraining. Note that PCA is already "transferable" to new datapoints in a way that most non-linear dimensionality reduction methods are not.*
> > >
> > > We refer the reviewer to the notion of transferability of GCNs as discussed in [vi], where the graphs are considered to be random instances sampled from an object called graphon. In this sense, graphs that are sampled at different resolutions retain an information structure that can be exploited **without re-training** in GCNs. Our experiments demonstrate this convincingly for VNNs over different-resolution datasets collected over the brain, where the brain surface can be thought of as a continuum in the spirit of a graphon object. We will clarify this in the paper.
> > >
> > >
> > > We note that there is no trivial way through which PCA can be transferred to new data points of a different dimensionality. For instance, PCA performed on the FTDC100 dataset is not practically meaningful for the FTDC300 or FTDC500 datasets. We reiterate that classical dimensionality reduction methods (linear or non-linear) do not have this property. So respectfully, we do not follow the reviewer’s argument that *‘PCA is already "transferable" to new datapoints’* in this context. **Further clarifications by the reviewer on this aspect will be much appreciated.**
> > >
> > > *vi. Ruiz, Luana et al. "Graphon neural networks and the transferability of graph neural networks." Advances in Neural Information Processing Systems 33 (2020): 1702-1712.*
> > >
> > > >*If it admits non-linearity then this has to be related to non-linear dimensionality reduction which is not taken up in this paper.*
> > >
> > > We clarify that we study VNN as a non-linear information processing architecture (including its stability and transferability properties and connections with PCA), whose impact and applications go well-beyond that possible for dimensionality reduction tasks.
> > >
> > > We also note that our experiments accommodate non-linear relationships for PCA as we use ‘rbf’ kernel after performing PCA as a baseline. Since VNNs and PCA exploit the same covariance matrix, we believe that our experiments provided a fair comparison between VNNs and PCA-based methods while accommodating non-linearity.

---

### Official Review · Reviewer_EVHZ · 2022-07-11

**Rating:** 7
**Confidence:** 3
**Soundness:** 3 good
**Presentation:** 4 excellent
**Contribution:** 3 good

**Summary:**

Motivated by similarities between principal components analysis (PCA) and graph convolutional filters in GNN, the authors introduce a new GNN architecture that uses sample covariance matrices as the graph representation of the data. They develop “coVariance” filters analogous to graph convolutional filters in GNNs, show that PCA is a special case of applying such filters, and propose a deep learning architecture based on “coVariance” filters they call “coVariance Neural Networks” (VNNs). Using perturbation theory for sample covariance matrices they theoretically establish the stability of VNNs to perturbations in the sample covariance matrix (in terms of number of samples). They then empirically evaluate VNN stability relative to PCA on synthetic and real world data, and show that VNNs can be used transferably on one multiresolution dataset.

**Questions:**

I would like to know more about model training time and computational cost in terms of covariance matrix size and number of samples.

**Limitations:**

I think the authors adequately address limitations in their work.


**Strengths And Weaknesses:**

Strengths:
- The work is original, making an interesting connection between GNNs and PCA and rigorously following this through the development, theoretical stability, and empirical validation of VNNs.
- The results are likely to be of interest to a broad audience, given the widespread use of PCA and prevalence of correlation matrices across disciplines.
- The theoretical stability analysis is sound and backed up by validation on synthetic and interesting real world neuroimaging data.
- The demonstration of multi-scale transferability is very cool, and goes beyond what PCA is capable of.
- The manuscript is clearly written and code is provided for replicability.

Weaknesses:
- I find the paper to have few weaknesses.

---

> ### Author Response · Authors · 2022-07-30
> **Response to Reviewer EVHZ**
>
> We thank the reviewer for their evaluation of our paper. The computational cost in any layer of VNN is determined by the cost of the convolution operation. Therefore, the computational cost for a covariance perceptron defined in (14) is given by $O(m^2 T F_{\sf in} F_{\sf out})$, where $T$ is the maximum number of filter taps in any filter in its associated filter bank and $m$ is the covariance matrix size. Moreover, there is a computation cost associated with the calculation of the covariance matrix, which is given by $O(mn^2)$, for $n$ number of samples when $n>m$. We also note that the factor of $m^2$ in $O(m^2 T F_{\sf in} F_{\sf out})$ is driven by the maximum density of the covariance matrix and therefore, could potentially be reduced by adopting sparse covariance estimation like approaches.
>
>
> In practice, for a VNN with 2 layers, 2 filter taps per layer and 44 features per dimensions, and trained over 100 epochs, the total training time is:
> * 22.86 seconds for FTDC100 (m=100)
> * 47.86 seconds for FTDC300 (m=300)
> * 89.73 seconds for FTDC500 (m=500)
>
> In addition, the aforementioned observations also help elucidate the contribution of transferability property of VNNs in enabling scalability to high resolution datasets by training the model on coarser/low resolution data.
>
> In the revised paper, we have briefly discussed the computational complexity in Remark 2 as well as the role of transferability in scalability of VNN.

---

### Official Review · Reviewer_TzhN · 2022-07-12

**Rating:** 7
**Confidence:** 4
**Soundness:** 4 excellent
**Presentation:** 3 good
**Contribution:** 4 excellent

**Summary:**

In this paper, the author first makes an observation that the filters of GNNs show similarities with principal component analysis (PCA), in which data is projected on the eigenspace of the covariance matrix. Then they proposed the covariance neural network (VNN) that operates on sample covariance matrices as graphs are motivated by this observation. They theoretically demonstrate VNN stability to perturbations in the covariance matrix, indicating a qualitative advantage over traditional PCA-based data analysis approaches that are prone to instability due to close eigenvalues and principal components. The author also makes several real work experiments to empirically prove their statement.

**Questions:**

How about the scalability of VNN? The experience of this paper is mainly focused on 'small' graphs. When it comes to large graphs, is it still computationally feasible? When VNN is used in large graphs, what will be the biggest challenge?

**Ethics Review Area:**

["I don’t know"]

**Limitations:**

There is no limitation and social impact in this paper. According to Neurips instructions, hope these two parts will appear in a future version.

**Strengths And Weaknesses:**


Strengths:
In this paper, the author links the coVariance filter with the graph convolution filter in GNN and proposes a DL architecture based on the coVariance filter. The computing covariance matrix for a large graph is always computation expensive, by alternatively computing graph Fourier transformation show potential for further GNN development.

The author also theoretically analyzes the stability of the covariance filter and covariance graph neural network and empirically evaluates VNNs for transferability and stability.


Weakness:

There is no related work section in this paper. Although those who study GNN are familiar with GNN, PCA, and graph Fourier transformation, it would be better to give an overview of different kinds of GNN and summaries them.

---

> ### Author Response · Authors · 2022-07-30
> **Response to Reviewer TzhN**
>
> We thank the reviewer for their valuable feedback. We have added an overview of GNNs that summarizes different GNN architectures and robust PCA as related work in Appendix E in the supplementary material. If the paper is accepted, we will add this as Subsection 1.3 in the introduction in the final version.
>
> The computational cost for a covariance perceptron defined in (14) is given by $O(m^2 T F_{\sf in} F_{\sf out})$, where $T$ is the maximum number of filter taps in any filter in its associated filter bank and $m$ is the covariance matrix size. Therefore, scalability to large covariance matrices is indeed the most challenging aspect due to increased computational complexity in terms of $m$. The transferability property of VNNs as illustrated by our experiments in Section 5.2 addresses the issue of scalability, where VNNs can be trained on a ‘coarser’ dataset first and then, the model is transferred to a higher dimensional/higher resolution dataset while retaining performance on the inference task. We also note that the factor of $m^2$ in $O(m^2 T F_{\sf in} F_{\sf out})$ is driven by the maximum density of the covariance matrix and therefore, could potentially be reduced by adopting sparse covariance estimation like approaches.
>
> We have added a brief discussion on the scalability of VNNs to large graphs as a potential limitation in the revised version of the paper (see Remark 2 in the revised manuscript). Our work does not have any negative social impacts.

---

### Meta-Review · Area_Chair_kTQE · 2022-08-25

**Recommendation:** Accept
**Confidence:** Certain

**Metareview:**

This paper proposes coVariance neural networks (VNN), which is a new architecture of graph neural network that is more robust to perturbations in covariance matrix. Most reviewers liked the new architecture as the intuition is clearly presented and the experiment results are interesting (in particular the results demonstrating multi-scale transferability). There are some concerns that this new architecture can be viewed as a more direct modification of GNN, I recommend the authors to clarify this relationship more clearly and emphasize the motivation.

**Award:**

No

---

### Decision · Program_Chairs · 2022-09-14

Accept